

# Simulating ice layer formation under the presence of preferential flow in layered snowpacks

Nander Wever[1,2], Sebastian Würzer[2], Charles Fierz[2], and Michael Lehning[1,2]

1 École Polytechnique Fédérale de Lausanne (EPFL), School of Architecture, Civil and Environmental Engineering, Lausanne, Switzerland.

WSL Institute for Snow and Avalanche Research SLF, Davos, Switzerland.

*Correspondence to:* Nander Wever (wever@slf.ch)

**Abstract.** For physics based snow cover models, simulating the formation of dense ice layers inside the snowpack has been a long time challenge. Their formation is considered to be tightly coupled to the presence of preferential flow, which is assumed to happen through flow fingering. Recent laboratory experiments and modelling techniques of liquid water flow in snow have advanced the understanding of conditions under which preferential flow paths or flow fingers form. We propose a modelling
approach in the one-dimensional, multi-layer snow cover model SNOWPACK for preferential flow that is based on a dual-domain approach. The pore space is divided into a part that represents matrix flow and a part that represents preferential flow. Richards equation is then solved for both domains. We found that preferential flow paths arriving at a layer transition in the snowpack may lead to ponding conditions. Subsequent refreezing then can form dense layers in the snowpack, that regularly exceed 700 kg m$^{-3}$. A comparison of simulated density profiles with bi-weekly snow profiles made at the Weissfluhjoch
measurement site at 2536 m altitude in the Eastern Swiss Alps for 16 snow seasons showed that several ice layers that were observed in the field could be reproduced. However many profiles remain challenging to simulate. The prediction of the early snowpack runoff also improved under the consideration of preferential flow. Our study suggests that a dual domain approach is able to describe the net effect of preferential flow on ice layer formation and liquid water flow in snow in one-dimensional, detailed, physics based snowpack models, without the need for a full multi-dimensional model.

## 1 Introduction

Ice layers form a marked microstructural transition inside the snowpack (*Fierz et al.*, 2009). Their formation is generally considered to be tightly coupled to the presence of preferential flow in snow (*Marsh and Woo*, 1984; *Pfeffer and Humphrey*, 1998; *Fierz et al.*, 2009). Despite their often small vertical extend, (thin) ice layers may have a profound impact on large scale processes in a snowpack, such as liquid water, heat and vapour flow (*Colbeck*, 1991; *Hammonds et al.*, 2015). Many fields 20 of study have addressed the issue of ice layers in snowpacks. Water may flow laterally over ice layers or crusts, reducing travel times and significantly impact catchment scale hydrology, although on the other hand, preferential flow in snow may promote vertical percolation instead (*Eiriksson et al.*, 2013). Recent studies have demonstrated that the increased melt on the Greenland Ice Sheet lead to changes in the firn structure, particularly through the formation of ice layers by percolating water in sub-freezing snow (*de la Peña et al.*, 2015). These ice layers can reach considerable vertical extends in the order of 1 m



(*Machguth et al.*, 2016) and may reduce the storage capacity of melt water in the firn by making access to deeper firn layers more difficult. Subsequent melt events may thus be accompanied by much more efficient runoff, due to lateral flow over these ice layers (*Pfeffer et al.*, 1991). Ice layers can also have a profound impact on microwave emission from snow covers, which is used in remote sensing retrieval algorithms (*Rees et al.*, 2010; *Roy et al.*, 2016). For rock stability of permafrost affected

regions, the presence of ice layers near the base of the snowpack as well as inside the snowpack was found to prevent liquid water from reaching joints in the rocks, thereby improving rock stability (*Phillips et al.*, 2016). Ice layers in snow covers also impact the access to food resources for wild life in snow covered areas (e.g., *Vikhamar-Schuler et al.* (2013)). Climate change projections of future increases in rain-on-snow events in high latitudes (*Ye et al.*, 2008), increased snow melt on ice sheets (*de la Peña et al.*, 2015) as well as more frequent melt events in alpine snowpacks (*Surfleet and Tullos*, 2013) show urgency to

be able to determine how these changes affect the snowpack microstructure in the future.

For 1D snow cover models, whether they are physics based or simple, it is notoriously difficult to simulate the formation of ice layers. This can be understood as most models do not consider preferential flow, which is a crucial transport mechanism to allow downward propagating water flow in sub-freezing snow. Liquid water can thereby reach areas in the snowpack where the cold content is large enough to refreeze the percolating melt water and form ice layers (*Humphrey et al.*, 2012). Early attempts

by *Colbeck* (1979) and *Marsh and Woo* (1985) to describe preferential flow in snowpacks never found widespread deployment. When considering snowpack runoff, neglecting preferential flow may be justified for describing seasonal runoff characteristics (*Wever et al.*, 2014a). However preferential flow may be crucial for understanding the response of a snow cover on short, sub-daily time scales, for example during rain-on-snow events (*Rössler et al.*, 2014; *Wever et al.*, 2014b; *Würzer et al.*, 2016a). Also for wet snow avalanche formation, the exact location at which liquid water starts ponding can influence snow stability

(*Wever et al.*, 2016) and considering preferential flow may be important for the exact timing of when weak layers are reached by water.

Recently, multi-dimensional snow cover models have been developed to simulate preferential flow (*Hirashima et al.*, 2014), but those models simplified and neglected several snowpack processes (i.e., snow settling, snow microstructure evolution), meaning that they are not yet applicable to natural snow covers. Furthermore, multi-dimensional snow cover models generally

require more computational power, making them unsuitable for large scale deployment. However, those multi-dimensional model developments provide crucial insights that allowed for a parametrisation of a dual domain approach for preferential flow for the 1D, physics based, detailed SNOWPACK model (*Bartelt and Lehning*, 2002; *Lehning et al.*, 2002a, b), which we present in this study.

## 2 Dual Domain Implementation

To simulate preferential flow, we apply a dual domain approach as schematically shown in Fig. 1. The pore space is subdivided into a part that is involved in preferential flow, and a part that is representing matrix flow (labelled 1 in Fig. 1). For the construction of the domains and the exchange processes between both domains, we exploit recent results from laboratory and model experiments as well as applying concepts from hydrological modelling. The water flow in the model is described for



both the matrix and preferential flow domain by solving Richards equation for both domains sequentially at the commonly used SNOWPACK time step of 15 min. After solving Richards equation for the matrix domain, the exchange of water between the matrix and preferential flow domain is determined and vice versa. If the pressure head exceeds the water entry pressure head of the layer below (labelled 2 in Fig. 1) water moves from matrix to preferential flow (labelled 3 in Fig. 1). If the saturation in the preferential flow path exceeds a threshold (labelled 4 in Fig. 1), water moves back to the matrix domain (labelled 5 in Fig. 1). Only the matrix part is allowed to undergo phase changes and ice layers form when water moves back from preferential flow to matrix flow and refreezes. Preferential flow remains always in the liquid phase. Refreezing of preferential flow water is mimicked by moving water from preferential flow to the matrix flow domain (labelled 6 in Fig. 1). Below, the water exchange processes are described in more detail.

## 2.1 Defining the Dual Domains

For the dual domain approach, the pore space is subdivided in a matrix and preferential flow domain (denoted with 1 in Fig. 1). For soils, the relative area involved in preferential flow is often found to be a function of the ratio of system influx rate over saturated hydraulic conductivity (*Glass et al.*, 1989a, b) for a given soil texture. In the experimental data on snow presented by *Katsushima et al.* (2013), a more pronounced dependence of the preferential flow area with grain size is found, rather than with the system influx rate. We illustrate their experimental results graphically in Fig. 2. Whereas the grain size shows a distinct pattern of smaller preferential flow area for larger grain sizes, the system influx rate showed a rather ambiguous pattern, where not always increased influx leads to a larger preferential flow area. The grain sizes used in their experiments span over typical ranges found in natural snow covers and this dependence is important to take into account. It also has to be noted that the infiltration rates in their experiments exceed typical values in natural conditions. We therefore decided to determine the dependence of preferential flow area on grain size using the lowest experimental infiltration rate only. A fit to this selection of the data (see Fig. 2) provides the following expression for the preferential flow area ($F$):

$$F = 0.0584 r_{\mathrm{g}}^{-1.109}, \tag{1}$$

where $F$ is the preferential flow area fraction (-), and $r_{\mathrm{g}}$ is the grain radius (mm). The matrix flow domain is accordingly defined as $(1 - F)$. For numerical stability, $F$ is limited between 0.01 and 0.90. Generally grain size increases over time in snow, and this may lead to a situation where the preferential flow area in the next SNOWPACK time step is reduced below the required one to accommodate for the liquid water present in the preferential flow domain. We therefore additionally ensure that $F$ is large enough to contain all present preferential flow water.

For both domains, the relationship between pressure head and liquid water content (LWC) is described by the van Genuchten parametrisation for snow as experimentally determined by *Yamaguchi et al.* (2012). For the matrix flow domain, the saturated LWC is scaled by $(1 - F)$ and for the preferential flow domain by $F$. Furthermore, we determine the residual water content for the matrix flow domain using the approach described in *Wever et al.* (2014a), while setting it to 0 for the preferential flow





domain. Saturated hydraulic conductivity is parametrised using the parametrisation for permeability proposed by *Calonne et al.* (2012).

## 2.2 Water Exchange Between Matrix and Preferential Flow Domain

All liquid water input (snow melt, rainfall, condensation) is added to the matrix flow domain. A prerequisite for the formation of an unstable wetting front (i.e., flow fingering) is that the system influx rate is below the saturated hydraulic conductivity of the medium (*DiCarlo*, 2013), which is generally fulfilled in snow (*Katsushima et al.*, 2013). In order to initiate preferential flow, we use the concept that preferential flow paths form when the pressure head in the matrix flow domain exceeds the water entry pressure of the layer below. This was found to be the case in laboratory experiments (*Katsushima et al.*, 2013; *Avanzi et al.*, 2015) and was successfully exploited in numerical modelling to initiate preferential flow (*Hirashima et al.*, 2014). The water entry pressure $h_{\mathrm{we}}$ (m) can be expressed as a function of grain size according to *Katsushima et al.* (2013):

$$h_{\mathrm{we}} = 0.0437 \, (2r_{\mathrm{g}}) + 0.01074 \,, \tag{2}$$

One important condition to reach the water entry pressure is water ponding on a microstructural transition inside the snowpack (*Hirashima et al.*, 2014; *Avanzi et al.*, 2015). This is denoted with 2 in Fig. 1. To achieve the high LWC value observed in experiments (*Avanzi et al.*, 2015), we use the geometric average to calculate the hydraulic conductivity between snow layers (*Wever et al.*, 2015). In our implementation, the amount of water in the matrix part in excess of the threshold corresponding to the water entry pressure of the layer below, is moved to the preferential flow domain in the layer below (denoted with 3 in Fig. 1). If afterwards the saturation (i.e., ratio of water volume to pore volume) in the layer in the matrix domain is higher than the saturation in the layer below in the preferential flow domain, the saturation is equalized by an equivalent water flow with the following approach. Equal saturation in a specific layer in the matrix domain and a layer in the preferential flow domain can be expressed as:

$$\frac{\theta_{\mathrm{m}} - \theta_{\mathrm{r,m}}}{\theta_{\mathrm{s,m}} - \theta_{\mathrm{r,m}}} = \frac{\theta_{\mathrm{p}} - \theta_{\mathrm{r,p}}}{\theta_{\mathrm{s,p}} - \theta_{\mathrm{r,p}}} \,, \tag{3}$$

Where the subscripts m and p denote the matrix and preferential flow domain, respectively, $\theta$ is the LWC (m$^3$ m$^{-3}$), $\theta_{\mathrm{r}}$ is the residual LWC (m$^3$ m$^{-3}$) and $\theta_{\mathrm{s}}$ is the saturated LWC (m$^3$ m$^{-3}$).

Given layer thicknesses $L_{\mathrm{m}}$ and $L_{\mathrm{p}}$ for the layers in the matrix flow and preferential flow domain, respectively, the total LWC in the matrix and preferential flow layer is defined as:

$$\theta_{\mathrm{tot}} = \theta_{\mathrm{m}} L_{\mathrm{m}} + \theta_{\mathrm{p}} L_{\mathrm{p}} \tag{4}$$

Under the requirement of an equal degree of saturation for a given total LWC, we can solve Eq. 3 for $\theta_{\mathrm{m}}$:

$$\theta_{\mathrm{m}} = -\frac{(\theta_{\mathrm{r,m}}\theta_{\mathrm{s,p}} - \theta_{\mathrm{r,p}}\theta_{\mathrm{s,m}}) \, L_{\mathrm{m}} + (\theta_{\mathrm{r,p}} - \theta_{\mathrm{s,p}}) \, \theta_{\mathrm{tot}}}{(\theta_{\mathrm{s,m}} - \theta_{\mathrm{r,m}}) \, L_{\mathrm{m}} + (\theta_{\mathrm{s,p}} - \theta_{\mathrm{r,p}}) \, L_{\mathrm{p}}} \,, \tag{5}$$

after which $\theta_{\mathrm{p}}$ can be found by applying Eq. 4.



Additionally, if the saturation in the matrix domain exceeded the saturation of the preferential flow domain in a snowpack layer, saturation is equalized using Eq. 5, with $L_\mathrm{m} = L_\mathrm{p}$. This is motivated by the fact that once snow is wet, no horizontal gradients in pressure head are expected to be present in a snow layer, and thus, following the Van Genuchten water retention curve, the saturation of the matrix domain is equal to the saturation of the preferential flow domain.

Conceptually, water will leave the preferential flow domain and enter the matrix domain if the pressure head inside the preferential flow domain exceeds the water entry pressure of the dry snow around the preferential flow path. This procedure was able to simulate water spreading on microstructural transitions in the multi-dimensional snow model by *Hirashima et al.* (2014). However, in our study, this approach rarely succeeded in forming ice layers, as the condition is rather seldom met. This fact can be interpreted in light of the physics behind preferential flow. It was demonstrated that an overshoot condition exists in

flow fingers, which means that the tip of a flow finger shows marked higher saturation than the tail. This flow behaviour cannot be described by Richards equation (*DiCarlo*, 2013), although Richards equation continues to provide a correct description above and below the wetting front. The reason why the condition worked in *Hirashima et al.* (2014) may be due to the fact that the simulations involved high water influx rates, much higher than experienced in natural snow covers. This would increase the amount of water accumulating on the capillary barrier when liquid water flow over the transition is slower than the water flux

arriving from above. In the absence of a solution for this problem (*DiCarlo*, 2013), we simply apply a threshold in saturation ($\Theta_\mathrm{th}$) of the preferential flow domain (denoted with 4 in Fig. 1). Once this threshold is exceeded, water will flow back to the matrix domain (denoted with 5 in Fig. 1). In our approach, we first move as much water as freezing capacity is available in the matrix domain. If after this approach the threshold is still exceeded, we additionally equalize the saturation in the specific layer in the matrix and preferential flow domain, using Eq. 5. For the lowest snow layer above the soil, the saturation is

always equalized between the matrix and preferential flow domain, regardless of whether the saturation threshold is exceeded or not. This suppressed spiky snowpack runoff behaviour. In soil layers, preferential flow is ignored by setting the hydraulic conductivity for the preferential flow domain to 0 and the preferential flow area to 2 %.

### 2.3 Refreezing Preferential Flow

In our approach, water in the preferential flow domain is not considered for phase changes. However, in reality preferential flow

is known to refreeze, even forming ice structures in the shape of flow fingers inside the snowpack (*Kattelmann*, 1985; *Marsh*, 1988; *Fierz et al.*, 2009; *Williams et al.*, 2000). For simplicity, we currently do not consider microstructural changes due to preferential flow, although they may have a strong effect on the water flow in snow. Grain growth and subsequent reduction of capillary forces as well as ice columns may increase the efficiency of the preferential flow paths considerably.

For the thermal effects, we first describe the heat flux between the preferential flow part and the matrix part by assuming a

pipe with radius $r$ (m) at melting temperature $T_0$ (K) in the middle of a 1 m$^2$ snowpack at temperature $T_\mathrm{e}$ (K) (see Fig. 3). If we assume that the horizontal temperature gradient inside the snowpack is linear, then the temperature $T_\mathrm{e}$ is found at a radius $R^*$ (m) such that surface areas $A_1$ (m$^2$) and $A_2$ (m$^2$) are equal. We can then approximate Fourier's law for heat flow for the





heat flux between the preferential flow and matrix domain ($Q_{H,\text{p}\rightarrow\text{m}}$, J m$^{-1}$ s$^{-1}$) as:

$$Q_{H,\text{p}\rightarrow\text{m}} = \kappa \frac{\partial T}{\partial x} \approx \kappa \frac{(T_\text{e} - T_0)}{\left(\sqrt{\frac{1+F}{2\pi}} - \sqrt{\frac{F}{\pi}}\right)} \tag{6}$$

The volumetric content that needs to be transferred from the preferential domain to the matrix domain (denoted with 6 in Fig. 1), in order to satisfy the refreezing capacity provided by the heat flux $Q_{H,\text{p}\rightarrow\text{m}}$ over the outer area of the preferential flow path can be subsequently expressed as:

$$\Delta\theta_{\text{w},\text{p}\rightarrow\text{m}} = N\pi F L_\text{e} Q_{H,\text{p}\rightarrow\text{m}} \Delta t \tag{7}$$

where $L_\text{e}$ is the latent heat associated with freezing ($3.34 \cdot 10^5$ J kg$^{-1}$) and $N$ is a factor describing the effect of multiple flow paths forming area $F$. Often, numerous flow paths can be identified per square meter of snowpack, as for example found in a field study by *McGurk and Marsh* (1995). They report a flow path density between roughly 100 to 300 per m$^2$. However, this number is not necessarily representative for the number of flow paths actively and concurrently transporting water, as often new preferential flow paths form in subsequent melt cycles (*Schneebeli*, 1995). *Albert et al.* (1999) found only 3 preferential flow paths per m$^2$ during the first wetting of a previously sub-freezing snowpack. When more flow paths are present, the energy exchange will be more efficient. Additionally, the gradients with the surrounding snow will be larger. We interpret $N$ a tuning parameter in the model related to the number of flow paths per m$^2$.

# 3 Data and Methods

## 3.1 Data

We simulate 16 subsequent snow seasons (2000-2015) for the Weissfluhjoch (WFJ) measurement site, located at 2536 m altitude in the Eastern Swiss Alps. For this site, a dataset of bi-weekly snow-profiles made in close vicinity (<25m) of the meteorological station used to drive the SNOWPACK model in this study is available (*WSL Institute for Snow and Avalanche Research SLF*, 2015; *Wever et al.*, 2015). The snow profiles contain information about grain size and type, judged by the observer using a magnifying glass, as well as snow density in sections of typically 20-50 cm height and snow temperature. Melt-freeze crusts (i.e., parts of the snowpack that have been wet and froze again), as well as ice layers are explicitly marked as such in the profiles. Ice lenses (i.e., non continuous ice layers) are not marked as an ice layer, but are reported in a separate remark. As subsequent snow profiles need to be made in undisturbed snow, they also sample spatial variability in addition to the temporal evolution. Furthermore, judging whether a specific layer is a crust or an ice layer is also partly subjective. This is also indicated in the data: sometimes the same layer is not identified similarly in subsequent snow profiles, although this may also indicate spatial variability. To account for spatial variability at the measurement site, we select the highest modelled dry snow density within a range of 20 cm above or below an observed ice layer, when comparing simulated and observed ice layers.

For validating the snowpack runoff simulated by the model, we use the snow lysimeter data from a 5 m$^2$ lysimeter, as described in *Wever et al.* (2014a). In that paper, it was discussed that a discrepancy between measured and modelled runoff is



particularly present at the beginning of the melt season, and involves the first ca. 5 % of seasonal snowpack runoff. Here, we consider the measured snowpack runoff for the period March 1 to May 31 only, and particularly focus on the first 20 mm w.e. runoff from the snowpack. This period corresponds to the onset of snowpack runoff, while preventing that the statistics are dominated by the main melt period. We additionally exclude lysimeter data from snow season 2000 and 2005 from the analysis,
given the suspicion of problems with the lysimeter in these seasons (*Wever et al.*, 2014a).

## 3.2 Methods

### 3.2.1 Model Setup

The simulation setup of the SNOWPACK model for WFJ is equal to the snow-height driven simulations in *Wever et al.* (2015), in which new snow fall amounts are determined from increases in measured snow height. This ensures a simulation that closely
follows the measured snow height, which will enable a correct comparison of simulated and observed snow profiles. Ice layers observed in the field can range from a few mm to a few cm and up to 1 m in firn on the Greenland Ice Sheet (*Fierz et al.*, 2009; *Machguth et al.*, 2016). To reduce computational costs, the SNOWPACK model applies an algorithm to merge elements when they exhibit similar properties. In default setting, this procedure typically maintains the layer spacing around 1.5 to 3 cm, except for certain special cases, like buried surface hoar or ice layers inside the snowpack. This means that in default setup, with a
typical layer spacing of 1.6 cm, the formation of ice layers is coupled to relatively thick layers compared to ice layers found in natural alpine snowpacks. Forming thinner ice layers requires less water and energy to refreeze. We therefore performed high resolution simulations where we lower the threshold above which no merge is allowed from 1.5 cm to 0.25 cm. Further, for the high resolution simulations, we initialize new snow layers during snowfall in steps of 0.5 cm, instead of the default value of 2 cm. This lead to a typical layer spacing of 0.45 cm. Results presented here are with the high resolution simulations, although
we discuss the performance of the default resolution as well. Simulations with matrix flow only took on average 2.3 min per simulated year to complete on a typical desktop PC. The dual domain approach, which requires solving Richards equation twice, increased the computation time to 8.0 min per year. The high resolution simulations, which we show here, took 71 min per year to complete.

Densities of ice layers in the field can vary over a wide range. For example, *Marsh and Woo* (1984) reports a range from 630
to 950 kg m$^{-3}$, which makes it ambiguous to determine above which threshold of modelled dry snow density a layer should be considered an ice layer. In the default setup, a layer with a dry snow density exceeding 700 kg m$^{-3}$ is considered an ice layer by the SNOWPACK model. However, we apply different thresholds here to verify the sensitivity of the choice of threshold on the results. Indeed, it may be that simulated layers cannot reach the density of observed, thin, ice layers due to their larger vertical extent in the model.





## 4  Results

### 4.1  Parameter Estimation

In the preferential flow formulation we propose, two tuning parameters are left: the threshold in saturation of the preferential flow domain ($\Theta_{\mathrm{th}}$), above which water will flow back to the matrix part, and a parameter related to the number of flow paths per m$^2$ ($N$). To determine an optimal set of parameters, a sensitivity study was carried out. For $\Theta_{\mathrm{th}}$, values from 0.02 to 0.16 in steps of 0.02 were used and for N, values 0, 0.2, 0.4, 0.6, 0.8, 1.0, 2.0, 3.0, 4.0, 5.0 were used.

Fig. 4 shows the probability of detection POD (i.e., the ratio of observed ice layers that are reproduced by the model over the total number of observed ice layers) for different thresholds that define a modelled ice layer, as a function of both tuning parameters. When ice layers are defined by higher densities, the POD decreases. Highest POD is achieved for no or minor freezing (i.e., small values for $N$), and a saturation in the preferential flow path around $0.1$. The non-linear relationship arises from the delicate balance of refreezing water that is not able to percolate deeper and ponding at microstructural transitions, required for water to move back to the matrix domain in order to freeze as an ice layer or lens. For snowpack runoff, highest scores in terms of $r^2$, RMSE or the arrival date of the first 20 mm w.e. are generally achieved with refreezing and low thresholds in saturation of the preferential flow domain (see Fig. 4). Both slow down the progression of preferential flow water. It seems difficult to find a set of parameters that will maximize both the reproduction of ice layers, as well as snowpack runoff simulations. Nevertheless, even with optimal settings for the formation of ice layers, the early stage of snowpack runoff (i.e., the passage time of the first 20 mm w.e. of runoff) is better reproduced than without considering preferential flow.

After executing all 80 SNOWPACK simulations for the sensitivity study, ranks were determined for the POD of ice layers, using 700 kg m$^{-3}$ as density threshold for ice layers, and the $r^2$ value for hourly snowpack runoff. The combination of both parameters that provides the lowest sum of the ranks for ice layer detection and snowpack runoff was considered the optimal combination of coefficients. This procedure gave $\Theta_{\mathrm{th}} = 0.1$ and $N = 0$ and $\Theta_{\mathrm{th}} = 0.08$ and $N = 0$ for the normal and high resolution simulations, respectively, as optimal combination of tuning parameters and this set of parameters will be used for the results. Interestingly, it implies that for ice layer formation, refreezing of preferential flow should be ignored (i.e., $N = 0$).

### 4.2  Example Snow Season

Fig. 5 illustrates the difference in simulated snow density between a simulation with only Richards equation and Richards equation including preferential flow at high resolution for snow season 2012. Similar figures for the other simulated snow seasons are shown in the Online Supplement. The overall density distribution is similar in both simulations, but only with preferential flow, ice layers are formed. The location is in good agreement with observations of ice layers and crusts observed in the snow profiles in the field. The distribution of liquid water is showing that the preferential flow is percolating ahead of the matrix flow. This partly is due to the absence of phase changes for water in preferential flow, but also that due to the lower area, and thereby lower value for $\theta_{\mathrm{s}}$, hydraulic conductivity increases faster with increasing LWC. Ponding at microstructural interfaces is occurring in both the matrix and the preferential flow domain and it marks the layers were water refreezes and forms ice layers. Solving Richards equation twice (for both domains) appears to be able to identify those layers.



In addition to preferential flow, ice layers can also form by surface processes. For example, rainfall in November 2003 in a sub-freezing snow cover formed an ice layer at the surface and this ice layer was subsequently observed during the rest of the 2004 snow season (see Fig. S5 in the Online Supplement). This layer is not reproduced in the SNOWPACK model, as the model did not recognize the precipitation as rainfall due to the low air temperature during the event. This layer has been 5 excluded in further analysis.

### 4.3 Density Profiles

Fig. 6 shows the observed snow density distribution in all snow profiles from snow seasons 2000-2015, typically representing vertical sections of 20-50 cm and sometimes smaller sections. The distribution of snow density for in these sections is well reproduced by the simulations, although the spread in simulated snow density is lower than the observed spread. All simulations 10 provide very similar snow density distributions. The $r^2$ value between observed and simulated density in the measurement sections is highest ($r^2$=0.74) for the simulations with Richards equation only and in *Wever et al.* (2015), it is shown that the temporal evolution and vertical distribution of snow density is in good agreement with measured snow density. With preferential flow, the $r^2$ value reduces to 0.71. Nevertheless, the simulations with preferential flow are maintaining the overall snowpack density profile generally well and the reduction in $r^2$ value may be attributed to the fact that calibration of snow 15 settling functions was not performed considering the preferential flow model. Another reason may be that with preferential flow, more water is moved downward and less water can refreeze in matrix flow in the upper snow layers. It may be argued that an underestimation of snow settling can be compensated for by an overestimation of refreezing water. In any case, the simulations with preferential flow stand out when looking at the highest snow density simulated within ±20 cm of an observed ice layer. In this case, much higher snow densities are found under consideration of preferential flow.

20 Fig. 7 shows the POD for different dry snow density thresholds that define an ice layer in the simulations. The POD decreases with increasing threshold from 0.44 for 400 and 500 kg m$^{-3}$ to 0.10 for a threshold of 800 kg m$^{-3}$ for the high resolution simulations. When comparing with field observations, it is important to note that it is not clear which density should be assigned to a layer that an observer would denote as an ice layer. The probability of null detection, which in this case is defined as the percentage of simulated profiles correctly simulating the absence of ice layers in the full profile is above 50 % for an ice layer 25 definition threshold of 600 kg m$^{-3}$ in high resolution simulations. In normal resolution simulations, the probability of null detection is higher. The bias, which is the ratio of the number of simulated ice layers over the number of observed ice layers, is generally below 1. This indicates a slight underestimation of the frequency of ice layers in both high and normal resolution simulations. It shows that our approach is neither largely overestimating nor underestimating the presence of ice layers inside the snowpack. The false alarm rate indicates that around half of the ice layers that are simulated do not find a correspondence 30 in the observed snow pits. The results illustrate the general difficulty of observing ice layers with often small vertical extent in the field and reproducing those ice layers in the model due to a delicate interaction between water flow and the ice matrix. However, the results also indicate that the model is able to capture a significant proportion of ice layers that formed in natural snowpacks, while maintaining the overall snowpack structure well.





### 4.4 Snowpack Runoff

In addition to ice layer formation, snowpack runoff is also strongly impacted by preferential flow. For rain-on-snow events at the WFJ measurement site, *Würzer et al.* (2016b) found an improvement in $r^2$ from $0.52$ for Richards equation only to $0.68$ for the dual domain approach. Interestingly, Fig. 8 shows that for the melt period, there is no consistent difference between $r^2$ values for daily and hourly snowpack runoff whether or not preferential flow is considered. On average both simulations have equal $r^2$ value of $0.81$ and $0.90$ for hourly and daily snowpack runoff, respectively. However, as already noted in *Wever et al.* (2014b), the effect of neglecting preferential flow on seasonal time scales may be very limited. In that study, particularly the first arrival of melt water was noticeably underestimated when only considering matrix flow with Richards equation. As illustrated in Fig. 9, the arrival time of the first 20 mm w.e. in the melt season is much better reproduced by the dual domain approach. The time difference between the arrival date of the first 20 mm w.e. changes from 7.7 days too late for the Richards equation model to 2.9 days too early in the dual domain approach. The standard deviation is also slightly smaller for the preferential flow formulation than for the matrix flow only. The fact that the standard deviation is smaller indicates that yearly variability between observed and simulated runoff is smaller and that the model is apparently able to better explain yearly variability. Generally, the absolute time difference between modelled and measured first 10 mm w.e. cumulative snowpack runoff is less than the absolute time difference for 20 mm w.e. cumulative runoff. This suggests that early season snowpack runoff from preferential flow is overestimated in the simulations. In *Würzer et al.* (2016b) additional confirmation is provided that the onset of snowpack runoff during rain-on-snow events is better reproduced by the dual domain approach.

### 5  Discussion

In the implementation of the dual domain approach, we attempted to stay close to a physics based process description. Laboratory experiments and multi-dimensional snowpack models have provided crucial insights in the preferential flow and water ponding processes. However, the number of quantitative experimental studies is still limited and many aspects may be refined in further studies. The model uses four criteria to specify the dual domain approach: (1) the area involved in preferential flow, (2) a condition to move water from matrix flow to preferential flow, (3) a condition to move water from preferential flow to matrix flow, (4) a condition describing the refreezing process of preferential flow. Two calibrating coefficients, related to criterion 3 and 4 were used to optimize the simulations.

The area involved in preferential flow (condition 1) is currently parametrised with grain size only. Given observations from soil physics (e.g., *Glass et al.* (1989b)), a dependence on the water influx rate is to be expected. Currently, laboratory settings, or field experiments with rainfall generators have generally large water input rates of typically 20 mm/hour or more (*Singh et al.*, 1997; *Katsushima et al.*, 2013; *Würzer et al.*, 2016b). It turns out to be difficult to have controlled, constant and spatially well distributed water input rates typically observed in nature (rainfall and melt rates of 1-5 mm/hour). The absence of studies at low water input rates makes the general validity of condition 1 we implemented uncertain.

We consider condition 2 to be a relatively well founded approach, as the role of water entry suction in forming preferential flow was clearly identified in laboratory experiments (*Katsushima et al.*, 2013) and turned out to be crucial in forming pref-



erential flow in multi-dimensional models in agreement with laboratory experiments (*Hirashima et al.*, 2014). However, also here, the exact parameterisation of water entry suction may be different for lower water influx rates.

Condition 3 may be one of the most uncertain ones. Understanding the LWC distribution in a preferential flow path cannot be achieved by the Richards equation (*DiCarlo*, 2013). The other issue is that infiltration in an initially dry porous medium is not

accurately described by Richards equation. We consider the assumption we made here that water will move from preferential flow to matrix flow based on the exceedance of a threshold in saturation one of the least supported by experimental results.

The refreezing of preferential flow (condition 4) is mainly limited by knowledge about the number of preferential flow paths that are actively transporting water, which in itself is dependent on snow cover conditions. Multi-dimensional snowpack models may help here to develop a better understanding of the heat exchange processes between preferential flow paths and

surrounding snow matrix, as a function of the number density of active preferential flow paths.

The term preferential flow can be interpreted ambiguously. Two phenomena are known to cause deviations from a matrix infiltration pattern: flow fingering and macropore flow. Here, we consider flow fingering purely as the result of instabilities of the wetting front, which can occur in porous media with a uniform pore space distribution. Generally the prerequisite for this effect (coarse grains and low infiltration rates) is fulfilled for snow. However, once flow fingering is occurring in snow,

microstructural changes of the snow grains in preferential flow paths will change the pore space distribution to a bimodal or multi-modal one. This has its equivalent in soils in, for example, worm holes, root channels and cracks. This effect is not considered in the dual domain approach we propose, although it may have a profound impact on the efficiency of preferential flow paths. Modifications to the parameters of the preferential flow domain can be imagined to better represent a multi-modal pore space distribution. On the other hand, snow microstructure inside and around preferential flow paths may not always

consist of an ice matrix where Richards equation would be a good description of water flow. However, a dual domain approach does not require both domains to be solved with Richards equation, and another description of water flow in the preferential flow domain may be more appropriate.

Our simulations have a relatively low reproduction success of observed ice layers. The sensitivity study has revealed that one factor is the delicate balance between refreezing and further percolation. This is expected to be particularly delicate in alpine

snow covers, where the cold content is low and the ice layers are often thin. For cold regions, for example the Greenland Ice Sheet, the abundance of ice layers observed in ice cores may be easier to reproduce in simulations. Microstructural transitions formed by summer melt-freeze crusts below cold new snow from the accumulation period are more easy to capture in simulations. In contrast with alpine snow covers, where the ground heat flux often maintains melting conditions at the snowpack base, firn temperatures are generally well below freezing, and create a large refreezing capacity.

Another factor contributing to the low probability of detection is the small vertical and sometimes small horizontal scale on which ice layer formation happens in alpine snowpacks, which is difficult to capture in simulations. A correct simulation of the snow microstructure is thereby a prerequisite for simulating ice layer formation, although it is difficult to achieve. As an example, buried surface hoar may provide a marked microstructural transition on which liquid water may pond and build ice layers. Whether or not the simulation is able to simulate correctly the burial of surface hoar, contributes to the failure or success

in reproducing ice layers. Such a failure or success will remain throughout the rest of the snow season.



# 6 Conclusions

We proposed a dual domain approach for modelling liquid water flow in snow, which separates the pore space in a part that is representing matrix flow and a part that is representing preferential flow. This dual domain approach for physics based snowpack models is able to simulate preferential flow paths such that a better agreement with the onset of snowpack runoff can

5  be achieved. The difference between the first modelled and measured 20 mm w.e. cumulative snowpack runoff decreased from approximately 8 days too late to 3 days too early. Furthermore, preferential flow ponding on microstructural transitions inside the snowpack and subsequent spreading in the matrix flow domain can simulate the otherwise lacking formation of dense layers by the model. Around 20 % of observed ice layers in the field over 16 snow seasons were correctly simulated by the model in the form of a layer exceeding a dry snow density of 700 kg m$^{-3}$. We showed that a dual domain approach is able to provide a

10  physics based description of preferential flow and ice layer formation that is corresponding to findings in laboratory and field experiments. However, the formulation has two parameters that were calibrated for this study. Although we do not resolve individual flow paths, as is done in multi-dimensional snowpack models, a dual domain approach can quantify the net effect of preferential flow on a snowpack in a 1-D snowpack model with much lower computational costs than multi-dimensional models and only marginally larger computational cost compared to 1-D non-multidomain models.

15  *Acknowledgements.* We thank the many employees of the WSL Institute for Snow and Avalanche Research SLF involved in taking the bi-weekly snow profiles at the Weissfluhjoch. Meteorological driving data for the SNOWPACK model as well as the bi-weekly snow profiles are accessible via doi 10.16904/1 (*WSL Institute for Snow and Avalanche Research SLF*, 2015-09-29) and 10.16904/2 (*WSL Institute for Snow and Avalanche Research SLF*, 2015), respectively. The SNOWPACK model is available under a LGPLv3 license at http://models.slf.ch. The version used in this study corresponds to revision 1028, of /branches/dev/pref_flow.



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



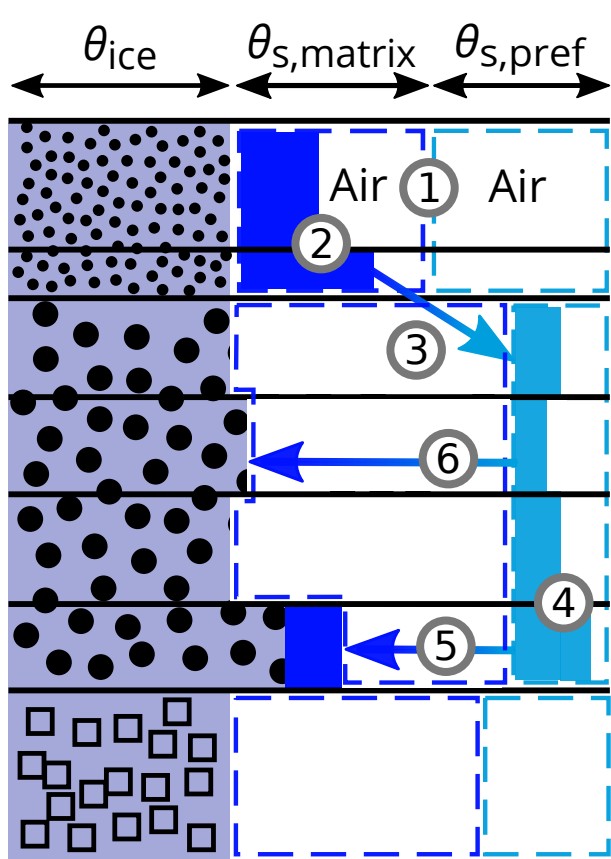

**Figure 1.** Schematic overview of the dual domain implementation for the SNOWPACK model, in which the pore space that can be occupied by liquid water is separated into a part for matrix flow ($\theta_{\mathrm{s,matrix}}$) and a part representing preferential flow ($\theta_{\mathrm{s,pref}}$). The numbers refer to processes described in the text.



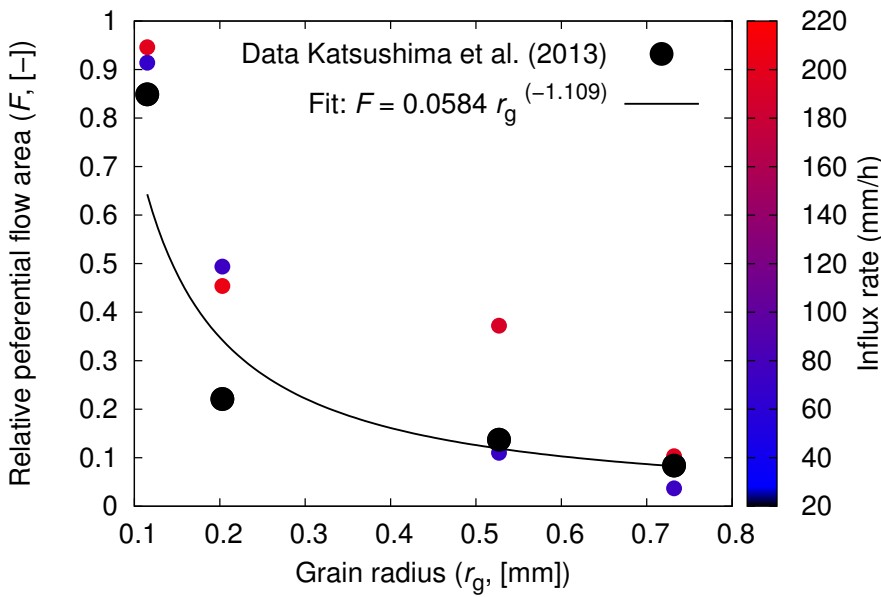

**Figure 2.** Relationship of the area involved in preferential flow as a function of grain radius. Data points represent laboratory experiments by *Katsushima et al.* (2013), presented quantitatively by *Hirashima et al.* (2014). Data points are coloured based on the water influx rate used in the experiments. The large black dots denote the data points used for determining the fit (solid line), corresponding to the lowest influx rate per grain size class.



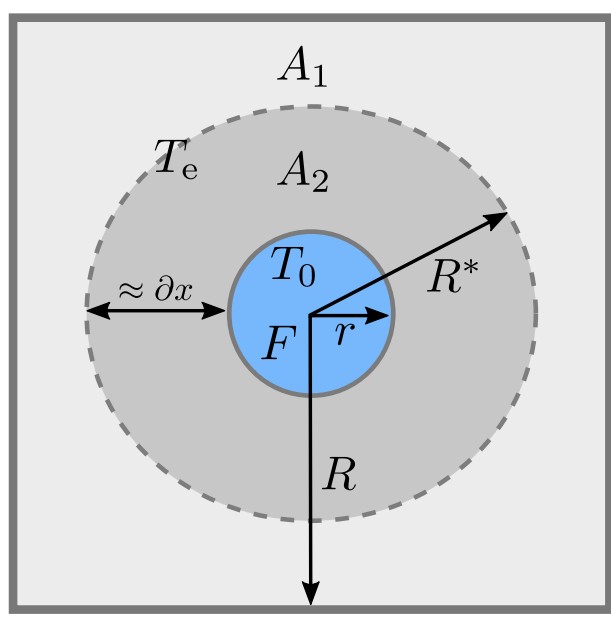

**Figure 3.** Schematic representation of a preferential flow path with radius $r$ and surface $F$ inside a 1 m$^2$ snowpack (i.e., $R = 0.5$ m), seen from above (not to scale), to approximate $\partial x$. The preferential flow path is assumed to be at melting temperature $T_0$, the rest of the snowpack at temperature $T_e$. $R^*$ is the radius such that surface areas $A_1$ and $A_2$ are equal. When assuming a linear temperature gradient, $T_e$ is found at distance $R^*$.





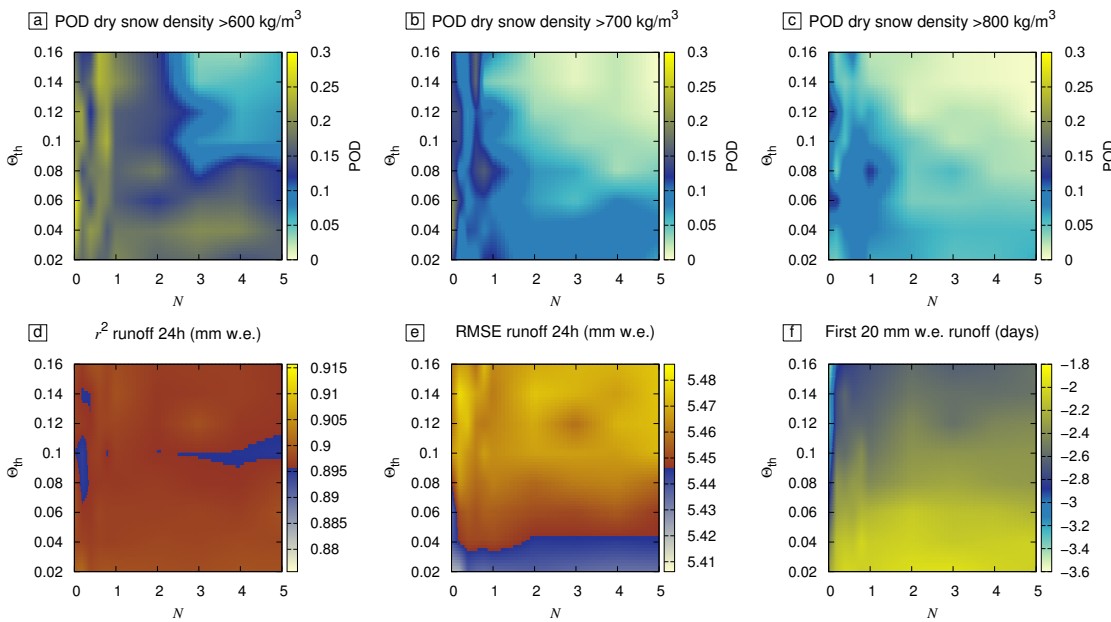

**Figure 4.** Interpolated results of the sensitivity study for the parameters $N$ and $\Theta_{\mathrm{th}}$ for the probability of detection (POD) when modelled dry snow density exceeds 600 kg m$^{-3}$ (a), 700 kg m$^{-3}$ (b), or 800 kg m$^{-3}$ (c) within 20 cm of the observed ice layer. For runoff, the $r^2$ for daily sums of runoff (d), the RMSE error for daily sums of runoff (e) and the number of days difference between modelled and measured passage of 20 mm w.e. since March, 1 of each snow season (f). The jump in colour scale from blue to red in (d) and (e) mark the score achieved with matrix flow only.





**Figure 5.** Dry snow density without considering preferential flow (a) and with preferential flow using high resolution simulations (b), validation with field observations (c) and liquid water content in the matrix and preferential flow domain for the simulation with preferential flow (d), for snow season 2012. In (c), modelled layers are shown when they are either a melt-freeze crust, or have a dry snow density exceeding 500 kg m$^{-3}$. For visibility, values of LWC in preferential flow below 0.1 % are ignored in (d).





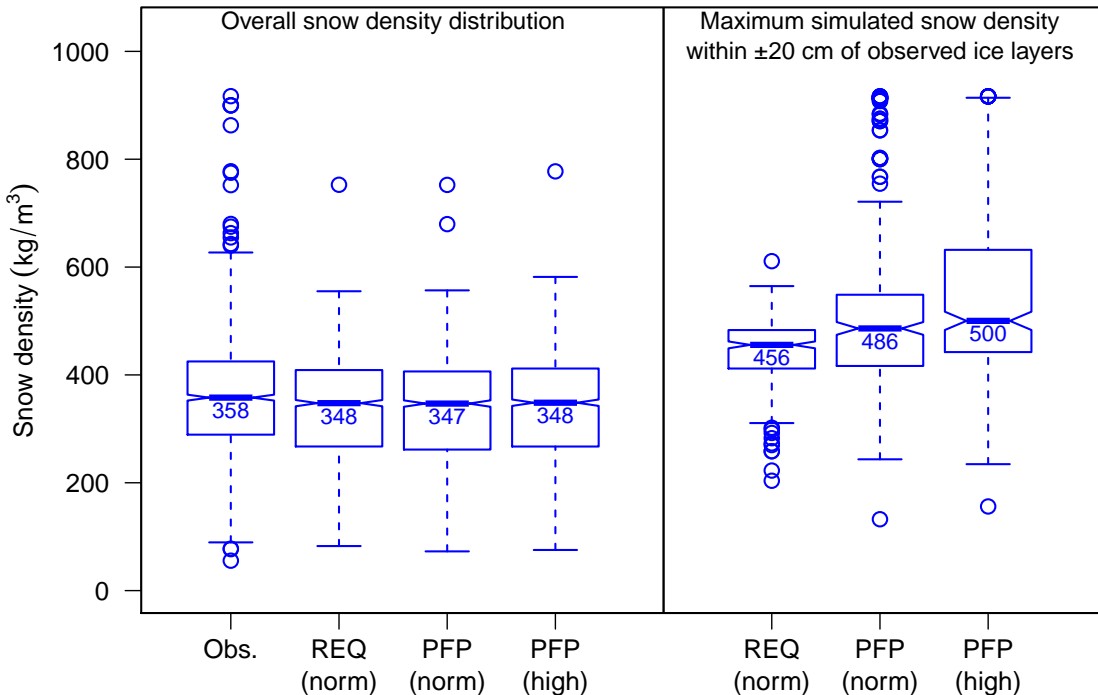

**Figure 6.** Box and whisker plot showing the distribution of snow density from observations (obs), simulations with Richards equation (REQ) and simulations with the dual domain approach to describe preferential flow (PFP). Boxes represent inter-quartile ranges (25th to 75th percentiles), thick horizontal bars in each box denote the median (50th percentile), its value shown directly below the bar. Whiskers (vertical lines and thin horizontal bars) represent the highest and lowest value within 1.5 times the inter-quartile range above the upper or below the lower quartile, respectively. Notches are drawn at ±1.58 times the inter-quartile range divided by the square root of the number of data points. Outliers are shown as individual dots.





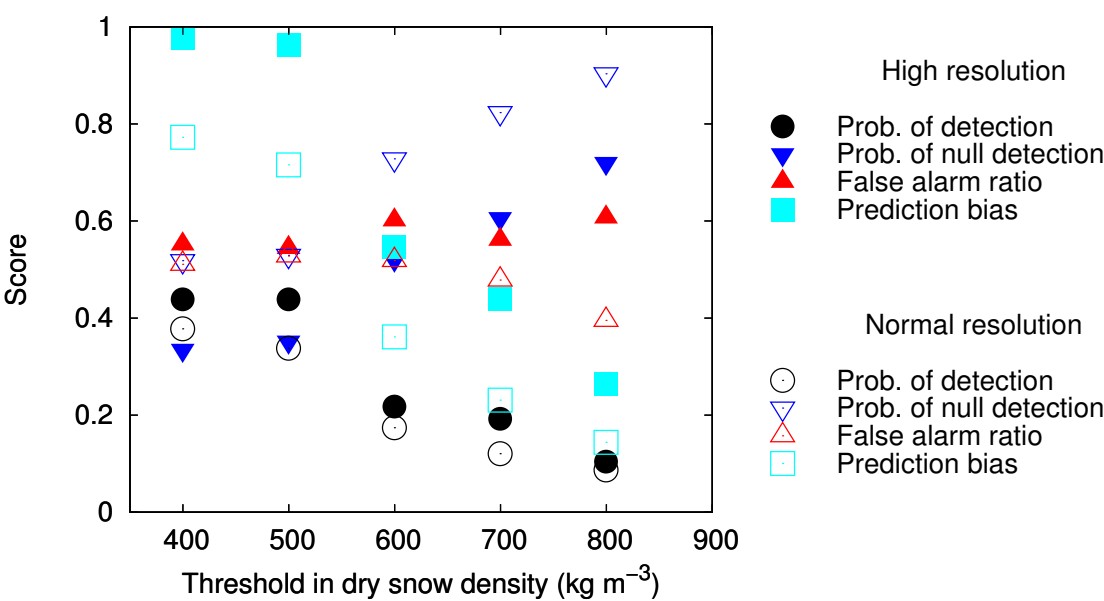

**Figure 7.** Contingency statistics as a function of threshold in dry snow density that defines an ice layer in the simulations, for both normal and high resolution simulations.





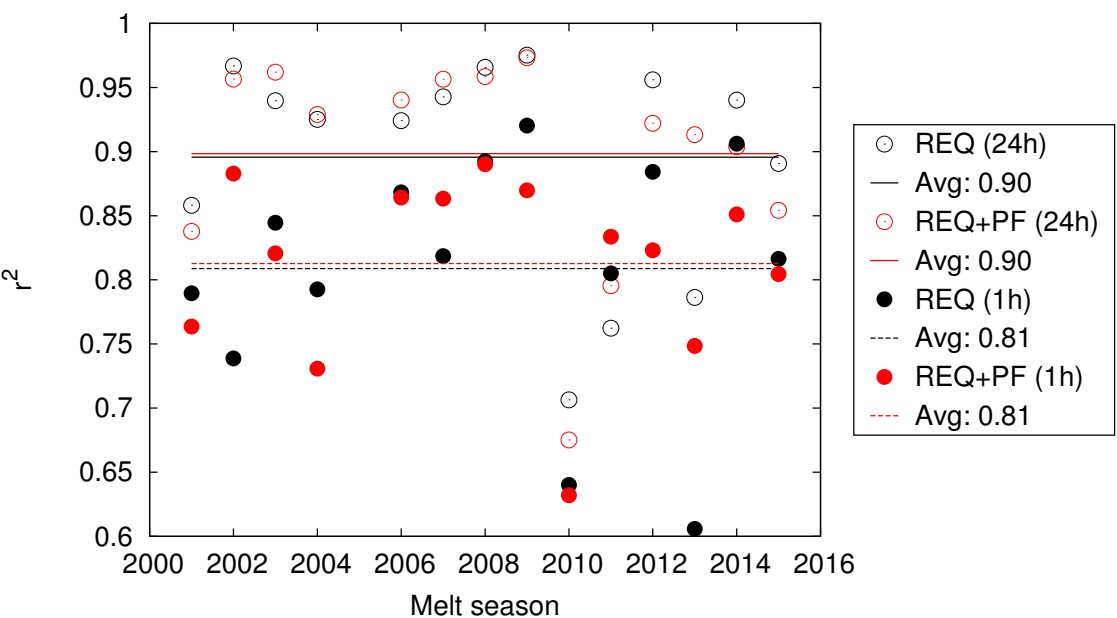

**Figure 8.** $r^2$ for both hourly and daily snowpack runoff over the period March 1 to May 31 for each snow season.

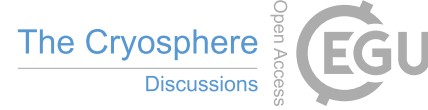



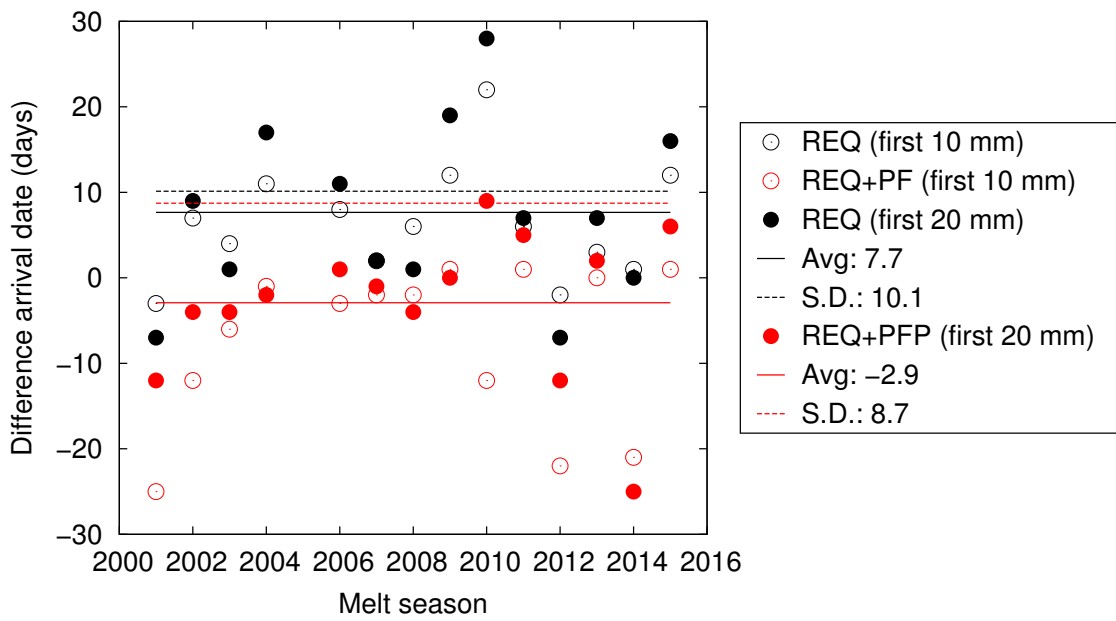

**Figure 9.** Difference (in days) between modelled and measured first 20 mm w.e. cumulative seasonal snowpack runoff (negative values denote modelled runoff is earlier than measured runoff).