# Peer review of "Simulating ice layer formation under the presence of preferential flow in layered snowpacks"

_The Cryosphere, 2016_

## Referee Comment (RC1) · Anonymous Referee #1 · 8 Sep 2016

Implementation of the preferential flow process into one-dimensional numerical snowpack model is valuable effort. Reproduction of ice layers in numerical snowpack model is also valuable. Furthermore, the dual domain approach is interesting idea. I appreciate the development of the new schemes to consider the preferential flow effect. On the other hand, considering the heterogeneous process in one dimensional model needs the various assumption, and it leads to the discrepancy between simulations and field observations. In the present state, this model still have the limitation of accuracy. In my opinion, achievement in this paper is new development of the concept to implement the preferential flow process in one-dimensional model with the purpose of reproduction of ice layers. The accuracy of this model is expected to be enhanced by cooperation with multi-dimensional model and laboratory experiment. In that context, the suggestion in the discussion section, laboratory experiment with small water input rates and heat

process simulation with multi-dimensional models, are important messages from this study. In my opinion, this paper is acceptable in the Cryosphere. I made lists following minor comments to make better contents of the paper.

minor comments

Introduction: Attempt to consider the effect of preferential flow in the numerical snow-pack model is also tried by Katsushima et al. (2009). The preferential flow process in their model is not physical base, but it is the start point of their experiment in Katsushima et al. (2013). I recommend to include following reference.

Katsushima, T., Kumakura, T., Takeuchi, Y., 2009. A multiple snow layer model including a parameterization of vertical water channel process in snowpack. Cold Regions Science Technology 59(2-3), 143-151.

P3 l22: In the present state, Equation (1) seems reasonable method to estimate the ratio of preferential flow area. However, this equation is too simplified and needs the improvement in the future. For example, considering only grain size is not sufficient. If author has any ideas of the experiment to improve this equation, I recommend to add the suggestion in this manuscript. It will be informative message for other researcher.

P3 l25 Usually, preferential flow path area get wider with time. Therefore, decrease in preferential flow area due to grain growth seems distant from actual process. However, in the dual domain simulation, if the decrease in preferential flow area due to grain growth leads the movement of water to matrix flow area, it can be considered as indirect expression for expansion of preferential flow area.

P8 L26 Fig.5 Can you add the detailed figures of snow temperature, density and water content focusing the beginning of March during the formation of ice layers? It helps the understanding why ice layer formed only the simulation with preferential flow.

p9L7 When the density data was counted, was the layer thickness considered? Also, Fig.6 show two figures, left one seems for all layers and right one seems data in specific

condition. However, in PFP simulation, the data near 900 kg/m3 existed in right figure despite it did not exist in left figure. It seems strange.

P9 L20 Figure 7 does not include the result of REQ. Ice layer may form even if the preferential flow is not considered depending on temperature and liquid water condition. Result of REQ had better be included in Fig. 7 to show the advantage of the consideration of preferential flow.

P10 6-7 No consist difference of r2 values considering preferential flow indicates that the matrix flow is predominant in this period. Thus, I guess most of snow was wet in this period. When enhancement in accuracy of runoff by considering preferential flow is discussed, information of snow stratigraphy should be included to show the ratio of dry snow, existence of ice layer and difference of grain size at layer boundary. Results of runoff simulation is discussed mainly in Würzer et al. So if their paper shows the snow stratigraphy as well as runoff simulation, it is not necessarily required in this paper.

P10-11 In the discussion section, descriptions '(1)P10L27-31, the absence of studies at low input rates makes the general validity of condition 1 we implemented uncertain (2)P10L6-9 Muti-dimensional snowpack models may help here to develop better understanding of the heat exchange processes between preferential flow paths and surrounding snow matrix, as a function of the number density of active preferential flow paths' are important messages. These suggestions provide the idea for valuable laboratory experiment and analysis using other model. If authors have other idea (e.g. the experiment to parameterize the process of transition from preferential flow to matrix flow.) and added in the manuscript, it will be welcomed as valuable information for reader studying wet snow.

---

## Referee Comment (RC2) · Anonymous Referee #2 · 8 Sep 2016

The authors have chosen to address a challenging topic and the results of this effort reflect the difficulty of simulating a heterogeneous, three dimensional, multi-process phenomenon in a 1-dimensional framework. While there is certainly much progress and future work to come in this area, the authors have presented a valuable analysis and initial framework that can be adapted and built upon in the future. As such, I am recommending that this work be accepted for publication in The Cryosphere, following some minor revisions that I believe will improve the quality and usefulness of the work.

Much of the framework relies on the interpretation of the results of Katsushima et al. (2013) as interpreted by Hirashima et al. (2014). They found that when water had reached the bottom of the profiles in their laboratory experiments, that the fractional area at 4 cm depth was smaller with larger snow grain size. The wetted fraction was interpreted here as the preferential flow fraction. As grain size decreased the wetted

fraction increased and this was interpreted as the preferential flow fraction being larger. However, as this extends to the finest grain size, it is stated that no preferential flow was observed, and indeed if slower matric flow was happening, the deepest wetted area would be larger and more uniform. If that is what happened, and matric flow had extended beyond the 4 cm depth, then we have no clear transition defined between the two flow regimes. I would like the authors to discuss the interpretation of the experiment on which Figure 2 is based.

Specific Comments:

P2 line 15: Although I would not expect a through presentation of the work of Colbeck (1979) or Marsh and Woo (1985), the authors have stated that these works were not widely adopted. A couple of sentences summarizing the main concepts presented in these studies, and any weaknesses that may have resulted in their lack of adoption, would help to inform the readers about the need for progress in this area of research.

P2 lines 19-21: I suspect that the specific application suggested in the example would require at least a 2-d model.

The dual-domain approach is a good starting place. Given that the authors point to three flow regimes (matric flow, flow fingering and macropore flow), it would seem appropriate to include multiple flow domains in the future; one step at a time.

P7: I suspect that there would be an interaction between the layer thicknesses (resolution) and the threshold of defining ice lenses / layers. A thicker layer would require more water equivalent to reach a given threshold density, and this would be harder to achieve in terms of water and energy transfer. A higher resolution or finer layers should enable higher threshold density values for defining ice layers.

P8 lines 7-17: This is an honest discussion of the performance issues but also shows that this methodology may be useful for future development.

P9 lines 1-5: Was the precipitation type generally known in the observations or was

this 2003 event a specific example of a known case in which the model misclassified the precipitation type? In any event, if it was known that the precipitation was rain, the model's diagnosis could be overwritten for this event, and if precipitation type was generally known, the model could be fed rainfall and snowfall separately. It would remove a source of uncertainty from the results. If the precipitation type was not generally known, but merely appeared to follow expected patterns save for the 2003 example, then it is not necessary to make any changes, given the size of the dataset.

P9 lines 7-13: I would expect the spread in simulated snow density to be somewhat smaller than observed with such a model, based on the fact that snow pits sample small spatial areas, and the presence of discontinuous ice layers and fingering may increase the number of samples necessary to obtain a reliable estimate of the mean density and its variability at a given depth. The PFP simulations in Figure 6 appears to capture the density distribution slightly better than the Richards equation alone. Have other statistical measures been explored as alternatives or in addition to $r^2$? A comparison of the mean and variance may show closer agreement with the PFP simulations. There is Willmott's index of agreement or one of the revised formulations. The objective is not to choose the statistic that makes the model appear better, but $r^2$ has been criticized as being insensitive to important factors of model performance.

In looking at Figures 8 and 9, there are differences in the performance of the REQ and REQ+PF models from year to year. It would be interesting to compare the conditions against a ranking of differences in $r^2$ (Fig. 8) and arrival date (Fig. 9). Are there specific snowpack or meteorological conditions that are correlated with the differences in performance between years? Knowledge of this may be useful for future model development.

Technical Comments:

P1 line 18: Change 'extend' to 'extent'

P1 line 20-23: Change 'Water may flow... instead (Eiriksson et al., 2013)' to 'Water

may flow laterally over ice layers or crusts, which reduces travel times and has a significant impact on catchment scale hydrology; alternatively, preferential flow in snow may promote vertical percolation instead (Eiriksson et al., 2013)'.

P1 line 23: This reads as if the reader is familiar with the increased melt on the Greenland Ice Sheet, which is likely to be true but, I would add a word or two to indicate the time frame.

P1 line 24: Change 'extends in' to 'extents, on'.

P2 line 6 (possibly elsewhere): I find the use of the term 'snow covers' to be awkward. I suggest the use of the term 'snowpack' or 'snowpacks'.

P2 line 15: I would change 'deployment' to 'adoption'.

P7 line 5: Change '... given the suspicion of problems...' to '... due to suspected problems...'.

P9 lines 8-9: Change '.... is well reproduced...' to '... is reproduced well...'.

---

## Author Comment (AC1) · 18 Oct 2016

**Response to Reviewer 1**

Implementation of the preferential flow process into one-dimensional numerical snowpack model is valuable effort. Reproduction of ice layers in numerical snowpack model is also valuable. Furthermore, the dual domain approach is interesting idea. I appreciate the development of the new schemes to consider the preferential flow effect. On the other hand, considering the heterogeneous process in one dimensional model needs the various assumption, and it leads to the discrepancy between simulations and field observations. In the present state, this model still have the limitation of accuracy. In my opinion, achievement in this paper is new development of the concept to implement the preferential flow process in one-dimensional model with the purpose of reproduction of ice layers. The accuracy of this model is expected to be enhanced by cooperation with multi-dimensional model and laboratory experiment. In that context, the suggestion in the discussion section, laboratory experiment with small water input rates and heat process simulation with multi-dimensional models, are important messages from this study. In my opinion, this paper is acceptable in the Cryosphere. I made lists following minor comments to make better contents of the paper.

*We thank the reviewer for his constructive comments and ideas to improve the manuscript. We agree that our approach is rather a starting point than a complete description of preferential flow and ice layer formation. Nevertheless, we also would like to stress that we think that the discrepancies between model and observation should not be attributed to model representation errors alone. Also inconsistencies and subjectiveness in snow pit observations, as well as inaccuracies in meteorological and snow lysimeter measurements to drive and verify the SNOWPACK model play an important role. Please find our detailed response to the issues raised by the reviewer below.*

**minor comments**

1. Introduction: Attempt to consider the effect of preferential flow in the numerical snowpack model is also tried by Katsushima et al. (2009). The preferential flow process in their model is not physical base, but it is the start point of their experiment in Katsushima et al. (2013). I recommend to include following reference. Katsushima, T., Kumakura, T., Takeuchi, Y., 2009. A multiple snow layer model including a parameterization of vertical water channel process in snowpack. Cold Regions Science Technology 59(2-3), 143-151.

   *We thank the reviewer for pointing our attention to this study, which certainly deserves citation in our manuscript. They used a similar concept to initiate preferential flow when ponding is occurring in the model domain. We will make appropriate references to this study when revising the manuscript.*

2. P3 l22: In the present state, Equation (1) seems reasonable method to estimate the ratio of preferential flow area. However, this equation is too simplified and needs the improvement in the future. For example, considering only grain size is not sufficient. If author has any ideas of the experiment to improve this equation, I recommend to add the suggestion in this manuscript. It will be informative message for other researcher.

   *We agree with the reviewer that Eq. 1 should be considered a preliminary result. Given the analogy between the ice matrix and soil, results from experiments with soil suggest that the preferential flow area in snow should most likely become a function of system influx rate. We think that repeating the experiments at low water input rates is an important step, although achieving low infiltration rates in a laboratory setting is generally challenging. We also think that confirmation of the absence of preferential flow for fine grains, as reported by Katsushima et al. (2013), needs to be acquired by increasing the sample size to exclude the possibility that the finger width is larger than the snow sample. This may*

*lead to the erroneous conclusion that the wetting of the complete snow sample shows the absence of preferential flow and only matrix flow is active. Please see also the major comment from Reviewer 2. We will amend the manuscript at this point.*

3. P3 l25 Usually, preferential flow path area get wider with time. Therefore, decrease in preferential flow area due to grain growth seems distant from actual process. However, in the dual domain simulation, if the decrease in preferential flow area due to grain growth leads the movement of water to matrix flow area, it can be considered as indirect expression for expansion of preferential flow area.

   *The situation described in P3L25 is not happening often. However, we wanted to describe our decision to limit the preferential flow area, instead of moving additional water from preferential flow to matrix flow if the preferential flow area decreases below the necessary area to accomodate all preferential flow water. While individual paths may increase with grain size, the data we used to establish the fit is showing that the total preferential flow area decreases with increasing grain size. We will amend the manuscript to report that the situation described in P3L25 happens seldom. We therefore also think that it should not be considered that this represents "an indirect expression for the expansion of the preferential flow area", as Reviewer 1 suggests. During the formation of preferential flow paths, paths indeed not only grow in length, but also in width. This is for example reported in Hirashima et al. (2014). However, this process is occurring on short time scales (typically within minutes/hours) when the preferential flow is developing towards a steady state. This widening is then likely not driven by grain growth, but by the non steady preferential flow path formation process. In the simulations, we aim to represent the steady state, particularly as SNOWPACK simulations are used to assess snow cover development on time scales from hours to a full season. Still, possible future revisions of Eq. 1 may be constructed for these kind of effects. On the other hand, it should maybe not be aimed for that 1-dimensional snowpack models with a dual domain descritpion describe the full dynamics of preferential flow paths, as long as the net effect is properly described. Such tasks may be more suited for full 3-dimensional snowpack models.*

4. P8 L26 Fig.5 Can you add the detailed figures of snow temperature, density and water content focusing the beginning of March during the formation of ice layers? It helps the understanding why ice layer formed only the simulation with preferential flow.

   *It is an interesting suggestion by the reviewer to show more of the processes occurring during the formation of ice layers. We will therefore include an additional figure in the manuscript, showing in more detail how snow density, grain size, snow temperature and liquid water flow interplay to form ice layers. This figure is shown in Fig. 1 in this document. The figure shows how preferential flow water (b) is percolating faster than matrix flow water (a), thereby reaching parts of the snowpack where the temperature is well below freezing (c). In (e) it can be seen that water accumulates on grain size differences between layers. By refreezing, melt-freeze crusts form (f) and when the density increases above 700 kg/m$^3$ (d), the model interprets the layer as an ice layer (f).*

5. p9L7 When the density data was counted, was the layer thickness considered? Also, Fig.6 show two figures, left one seems for all layers and right one seems data in specific condition. However, in PFP simulation, the data near 900 kg/m3 existed in right figure despite it did not exist in left figure. It seems strange.

   *The left hand side of Fig. 6 shows density for the segments as measured by the observer. These segments sample vertically about 30 cm of the snow cover at a time. The simulated snow profiles are aggregated to the same segments as measured by the observer, and then we show the average density over this segment. So indeed, we considered the layer thickness when the density data was processed. This is a comparison*

*of how the simulated density distribution agrees with the observed one. The problem with this analysis is that ice layers are thinner than the typical segments used by the observer, and are not sampled as such. Sampling the density of the actual ice layers is rather complicated (e.g., Watts et al. (2016)) and this is not done during the regular snow profiles at WFJ. Therefore, the right hand side of Fig. 6 shows the highest modelled density in a model layer, which was found within 20 cm of an observed ice layer. A model layer has a typical vertical extent of less than 2 cm. In this case, no density information is available from the observation at this level of detail, and the simulated snow density cannot be verified by the measurements. Apparently, our presentation of the analysis is causing confusion at this point and we will revise the manuscript and the figure caption to provide a better description of what is depicted in Fig. 6.*

6. P9 L20 Figure 7 does not include the result of REQ. Ice layer may form even if the preferential flow is not considered depending on temperature and liquid water condition. Result of REQ had better be included in Fig. 7 to show the advantage of the consideration of preferential flow.
*We revised the figure. Please see Fig. 2 in this document. For clarity, we only show the two most important statistics for simulations with Richards equation only. The statistics for the probability of detection shows that using Richards equation only, no ice layers with a dry snow density above 600-700 kg/m$^3$ are reproduced by the model. The prediction bias shows that high density layers are predicted much less frequent compared to how often they are found in the observations. Including preferential flow in the model, is clearly improving both the probability of detection as well as the prediction bias. We will amend the manuscript to include the revised figure.*

7. P10 6-7 No consist difference of r2 values considering preferential flow indicates that the matrix flow is predominant in this period. Thus, I guess most of snow was wet in this period. When enhancement in accuracy of runoff by considering preferential flow is discussed, information of snow stratigraphy should be included to show the ratio of dry snow, existence of ice layer and difference of grain size at layer boundary. Results of runoff simulation is discussed mainly in Würzer et al. So if their paper shows the snow stratigraphy as well as runoff simulation, it is not necessarily required in this paper.
*The manuscript by Würzer et al. (2016) shows the effect of preferential flow on short time scales during rain-on-snow events, which is a specific type of event. We also felt the need to show how the preferential flow formulation simulates snowpack runoff on seasonal time scales, not only during specific events. Therefore, we decided to include snowpack runoff analysis in the manuscript and we prefer to keep it here. Concerning the explanation of the year-to-year variability, we provide here the same response as to Reviewer 2, who raises a similar issue: it certainly is an interesting suggestion to try to explain year-to-year differences in performance of snowpack runoff simulations. However, we did not find a statistically significant correlation between, for example, the $r^2$ value or arrival date and the number of ice layers or the number of jumps in grain size or hardness observed in the profiles. We tested both linear correlation using Pearson correlation as well as rank correlation coefficients (Spearman, Kendall). One issue is that, as Reviewer 2 points out, meteorological conditions also vary from year to year. In some melt seasons, percolation speeds through the snowpack are high, in others it takes a few weeks for the whole snowpack to become wet. These differences arise from weather patterns: in some years warm weather prevails for several weeks in the melt season, leading to a quick wetting of the snowpack, whereas in other years, snow melt periods are interrupted by periods with colder weather and new snowfall amounts. We therefore also analyzed the maximum difference in the relative part of the snowpack that consists of melt forms in the observed profiles, between two subsequent snow profiles. This is an indication of the progress of the melt water front inside the snowpack. However, also this did not reveal any*

*statistically significant information. We think that ultimately, there are many factors contributing to high or low $r^2$ values or good or poor estimates of the arrival date: warming rate of the snowpack, presence of capillary barriers and ice layers that may trigger preferential flow, errors in meteorological measurements and errors in snow lysimeter measurements. The snow lysimeter at the Weissfluhjoch measurement site has a surface area of 5 $m^2$. As an illustration, Fig. 2 in Kattelmann (2000) shows that with this size, the variation coefficient is still quite large, and that an area of at least 10 $m^2$ may be required to more accurately capture snowpack runoff. Furthermore, inconsistencies in the bi-weekly profiles are present due to the subjective component of judging grain size and shape, as well as the fact that multiple observers are responsible for the snow profiles. All these factors make the analysis of factors contributing to high or low model performance difficult. We will discuss this in the revised manuscript.*

8. P10-11 In the discussion section, descriptions '(1) P10L27-31, the absence of studies at low input rates makes the general validity of condition 1 we implemented uncertain (2) P10L6-9 Muti-dimensional snowpack models may help here to develop better understanding of the heat exchange processes between preferential flow paths and surrounding snow matrix, as a function of the number density of active preferential flow paths' are important messages. These suggestions provide the idea for valuable laboratory experiment and analysis using other model. If authors have other idea (e.g. the experiment to parameterize the process of transition from preferential flow to matrix flow.) and added in the manuscript, it will be welcomed as valuable information for reader studying wet snow.

*We think the most crucial step is to investigate and quantify the heat exchange between preferential flow paths and the surrounding ice matrix, as a function of preferential flow area and the number of preferential flow paths. We expect that the vertical percolation speed of the preferential flow fingers is slowed down when percolating cold snow where considerable refreeze may take place. We think that a deeper investigation requires laboratory experiments with low water input rates (this may be difficult to achieve), high resolution temperature measurements (probably using infra-red photography) as well as dye-tracer to follow the wetting of the snowpack. Related to point 2, larger sample sizes may be required to prevent preferential flow finger width exceeding the sample size. We additionally think that using multi-dimensional models to simulate laboratory experiments helps to quantify the heat and water flow and to verify to what extent formulations of the heat and water exchange between preferential flow paths and the surrounding matrix reproduce the observations in laboratory experiments. For example, when a preferential flow path hits a microstructural transition, it starts spreading over the interface at some point. This is often explained via the water entry suction, yet the exceedance of the water entry suction in the preferential flow path in our simulations was not occurring often enough to successfully reproduce ice layers. Detailed laboratory and numerical studies using multi-dimensional models may assess the liquid water content distribution inside a preferential flow path. However, here it is important to note that some studies suggest that the water flow and wetness distribution inside a preferential flow path can probably not be described by Richards equation (DiCarlo, 2013). Ultimately, formulations for heat and water exchange between preferential flow paths and the surrounding matrix can then be incorporated in a model framework for 1-dimensional snowpack models, such as the one which we propose in this study. We will amend the manuscript at this point.*

[Figure]

Figure 1: LWC in matrix flow (a), LWC in preferential flow (b), snow temperature (c), snow density (d), grain radius (e) and grain shape (f), depicting a detail of Fig. 5 in the original manuscript. Only the upper part of the snowpack is shown for the period 27 February to 15 March. In (c), snow at melting temperature is coloured black to highlight wet parts and in (f), ice formations are defined as modelled dry snow density exceeding 700 kg/m$^3$.

[Figure]

Figure 2: Contingency statistics as a function of threshold in dry snow density that defines an ice layer in the simulations, for both normal and high resolution simulations including preferential flow (REQ+PF) and normal resolution simulations using Richards equation only (REQ).

**References**

DiCarlo, D. A. (2013), Stability of gravity-driven multiphase flow in porous media: 40 years of advancements, *Water Resour. Res.*, *49*(8), 4531–4544, doi:10.1002/wrcr.20359.

Hirashima, H., S. Yamaguchi, and T. Katsushima (2014), A multi-dimensional water transport model to reproduce preferential flow in the snowpack, *Cold Reg. Sci. Technol.*, *108*, 80–90, doi:10.1016/j.coldregions.2014.09.004.

Katsushima, T., S. Yamaguchi, T. Kumakura, and A. Sato (2013), Experimental analysis of preferential flow in dry snowpack, *Cold Reg. Sci. Technol.*, *85*, 206–216, doi:10.1016/j.coldregions.2012.09.012.

Kattelmann, R. (2000), Snowmelt lysimeters in the evaluation of snowmelt models, *Ann. Glaciol.*, *31*(1), 406–410, doi:10.3189/172756400781820048.

Watts, T., N. Rutter, P. Toose, C. Derksen, M. Sandells, and J. Woodward (2016), Brief communication: Improved measurement of ice layer density in seasonal snowpacks, *Cryosphere*, *10*(5), 2069–2074, doi:10.5194/tc-10-2069-2016.

Würzer, S., N. Wever, R. Juras, M. Lehning, and T. Jonas (2016), Modeling liquid water transport in snow under rain-on-snow conditions – considering preferential flow, *Hydrol. Earth Syst. Sci. Discuss.*, doi:10.5194/hess-2016-351.

---

## Author Response (AR1)

**Response to reviews**

Please find below our point-by-point response to all issues raised by both reviewers. The responses are the same as we uploaded to the online discussion. However, we amended here in which line numbers of the revised manuscript changes have been made.

**Response to Reviewer 1**

Implementation of the preferential flow process into one-dimensional numerical snowpack model is valuable effort. Reproduction of ice layers in numerical snowpack model is also valuable. Furthermore, the dual domain approach is interesting idea. I appreciate the development of the new schemes to consider the preferential flow effect. On the other hand, considering the heterogeneous process in one dimensional model needs the various assumption, and it leads to the discrepancy between simulations and field observations. In the present state, this model still have the limitation of accuracy. In my opinion, achievement in this paper is new development of the concept to implement the preferential flow process in one-dimensional model with the purpose of reproduction of ice layers. The accuracy of this model is expected to be enhanced by cooperation with multi-dimensional model and laboratory experiment. In that context, the suggestion in the discussion section, laboratory experiment with small water input rates and heat process simulation with multi-dimensional models, are important messages from this study. In my opinion, this paper is acceptable in the Cryosphere. I made lists following minor comments to make better contents of the paper.

We thank the reviewer for his constructive comments and ideas to improve the manuscript. We agree that our approach is rather a starting point than a complete description of preferential flow and ice layer formation. Nevertheless, we also would like to stress that we think that the discrepancies between model and observation should not be attributed to model representation errors alone. Also inconsistencies and subjectiveness in snow pit observations, as well as inaccuracies in meteorological and snow lysimeter measurements to drive and verify the SNOWPACK model play an important role. Please find our detailed response to the issues raised by the reviewer below.

**minor comments**

 Introduction: Attempt to consider the effect of preferential flow in the numerical snowpack model is also tried by Katsushima et al. (2009). The preferential flow process in their model is not physical base, but it is the start point of their experiment in Katsushima et al. (2013). I recommend to include following reference. Katsushima, T., Kumakura, T., Takeuchi, Y., 2009. A multiple snow layer model including a parameterization of vertical water channel process in snowpack. Cold Regions Science Technology 59(2-3), 143-151.

We thank the reviewer for pointing our attention to this study, which certainly deserves citation in our manuscript. They used a similar concept to initiate preferential flow when ponding is occurring in the model domain. We made appropriate references to this study when revising the manuscript, see P2, L21,28 and P4, L21.

2. P3 122: In the present state, Equation (1) seems reasonable method to estimate the ratio of preferential flow area. However, this equation is too simplified and needs the improvement in the future. For example, considering only grain size is not sufficient. If author has any ideas of the experiment to improve this equation, I recommend to add the suggestion in this manuscript. It will be informative message for other researcher.

We agree with the reviewer that Eq. 1 should be considered a preliminary result. Given the analogy between the ice matrix and soil, results from experiments with soil suggest that the preferential flow area in snow should most likely become a function of system influx rate. We think that repeating the experiments at low water input rates is an important step, although achieving low infiltration rates in a laboratory setting is generally challenging. We also think that confirmation of the absence of preferential flow for fine grains, as reported by Katsushima et al. (2013), needs to be acquired by increasing the sample size to exclude the possibility that the finger width is larger than the snow sample. This may lead to the erroneous conclusion that the wetting of the complete snow sample shows the absence of preferential flow and only matrix flow is active. Please see also the major comment from Reviewer 2. We amended the manuscript at this point: on P4, L2-3 and on P12, L17-21, we discuss now that preferential flow was not observed for fine grains and that larger snow samples may be required to confirm the absence of preferential flow for fine grained snow. Our suggestion that experiments at low input rates are probably needed to refine Eq. 1 was already mentioned in the original manuscript and can be found back in the revised manuscript on P12, L12-17.

3. P3 125 Usually, preferential flow path area get wider with time. Therefore, decrease in preferential flow area due to grain growth seems distant from actual process. However, in the dual domain simulation, if the decrease in preferential flow area due to grain growth leads the movement of water to matrix flow area, it can be considered as indirect expression for expansion of preferential flow area.

The situation described in P3, L25 in the original manuscript is not happening often. However, we wanted to describe our decision to limit the preferential flow area, instead of moving additional water from preferential flow to matrix flow if the preferential flow area decreases below the necessary area to accommodate all preferential flow water. While individual paths may increase with grain size, the data we used to establish the fit is showing that the total preferential flow area decreases with increasing grain size. We amended the manuscript to report that this situation happens seldom, see P4, L6. We therefore also think that it should not be considered that this represents "an indirect expression for the expansion of the preferential flow area", as Reviewer 1 suggests. During the formation of preferential flow paths, paths indeed not only grow in length, but also in width. This is for example reported in Hirashima et al. (2014). However, this process is occurring on short time scales (typically within minutes/hours) when the preferential flow is developing towards a steady state. This widening is then likely not driven by grain growth, but by the non steady preferential flow path formation process. In the simulations, we aim to represent the steady state, particularly as SNOWPACK simulations are used to assess snow cover development on time scales from hours to a full season. Still, possible future revisions of Eq. 1 may be constructed for these kind of effects. On the other hand, it should maybe not be aimed for that 1dimensional snowpack models with a dual domain description describe the full dynamics of preferential flow paths, as long as the net effect is properly described. Such tasks may be more suited for full 3dimensional snowpack models.

4. P8 L26 Fig.5 Can you add the detailed figures of snow temperature, density and water content focusing the beginning of March during the formation of ice layers? It helps the understanding why ice layer formed only the simulation with preferential flow.

It is an interesting suggestion by the reviewer to show more of the processes occurring during the formation of ice layers. We included an additional figure, Fig. 6 in the revised manuscript, showing in more detail how snow density, grain size, snow temperature and liquid water flow interplay to form ice layers. This figure shows how preferential flow water (b) is percolating faster than matrix flow water (a), thereby reaching parts of the snowpack where the temperature is well below freezing (c). In (e) it can be seen that water accumulates on grain size differences between layers. By refreezing, melt-freeze crusts form (f) and when the density increases above 700 kg/m3 (d), the model interprets the layer as an ice layer (f).

5. p9L7 When the density data was counted, was the layer thickness considered? Also, Fig.6 show two figures, left one seems for all layers and right one seems data in specific condition. However, in PFP simulation, the data near 900 kg/m3 existed in right figure despite it did not exist in left figure. It seems strange.

The left hand side of Fig. 6 in the original manuscript shows density for the segments as measured by the observer. These segments sample vertically about 30 cm of the snow cover at a time. The simulated snow profiles are aggregated to the same segments as measured by the observer, and then we show the average density over this segment. So indeed, we considered the layer thickness when the density data was processed. This is a comparison of how the simulated density distribution agrees with the observed one. The problem with this analysis is that ice layers are thinner than the typical segments used by the observer, and are not sampled as such. Sampling the density of the actual ice layers is rather complicated (e.g., Watts et al. (2016)) and this is not done during the regular snow profiles at WFJ. Therefore, the right hand side of Fig. 6 in the original manuscript shows the highest modelled density in a model layer, which was found within 20 cm of an observed ice layer. A model layer has a typical vertical extent of less than 2 cm. In this case, no density information is available from the observation at this level of detail, and the simulated snow density cannot be verified by the measurements. Apparently, our presentation of the analysis was causing confusion at this point and we revised the manuscript, see P10, L18-23, as well as the figure caption (see Fig. 7 in the revised manuscript).

- 6. P9 L20 Figure 7 does not include the result of REQ. Ice layer may form even if the preferential flow is not considered depending on temperature and liquid water condition. Result of REQ had better be included in Fig. 7 to show the advantage of the consideration of preferential flow. We revised the figure, which is now Fig. 8 in the revised manuscript. For clarity, we only show the two most important statistics for simulations with Richards equation only. The statistics for the probability of detection shows that using Richards equation only, no ice layers with a dry snow density above 600-700 kg/m3 are reproduced by the model. The prediction bias shows that high density layers are predicted much less frequent compared to how often they are found in the observations. Including preferential flow in the model, is clearly improving both the probability of detection as well as the prediction bias. We amended the manuscript to discuss the results for simulations using Richards equation only, see P11, L4-8.
- 7. P10 6-7 No consist difference of r2 values considering preferential flow indicates that the matrix flow is predominant in this period. Thus, I guess most of snow was wet in this period. When enhancement in accuracy of runoff by considering preferential flow is discussed, information of snow stratigraphy should be included to show the ratio of dry snow, existence of ice layer and difference of grain size at layer boundary. Results of runoff simulation is discussed mainly in Würzer et al. So if their paper shows the snow stratigraphy as well as runoff simulation, it is not necessarily required in this paper.

The manuscript by Würzer et al. (2016) shows the effect of preferential flow on short time scales during rain-on-snow events, which is a specific type of event. We also felt the need to show how the preferential flow formulation simulates snowpack runoff on seasonal time scales, not only during specific events. Therefore, we decided to include snowpack runoff analysis in the manuscript and we prefer to keep it here. Concerning the explanation of the year-to-year variability, we provide here the same response as to Reviewer 2, who raises a similar issue: it certainly is an interesting suggestion to try to explain year-toyear differences in performance of snowpack runoff simulations. However, we did not find a statistically significant correlation between, for example, the  $r^2$  value or arrival date and the number of ice layers or the number of jumps in grain size or hardness observed in the profiles. We tested both linear correlation using Pearson correlation as well as rank correlation coefficients (Spearman, Kendall). One issue is that, as Reviewer 2 points out, meteorological conditions also vary from year to year. In some melt seasons, percolation speeds through the snowpack are high, in others it takes a few weeks for the whole snowpack to become wet. These differences arise from weather patterns: in some years warm weather prevails for several weeks in the melt season, leading to a quick wetting of the snowpack, whereas in other years, snow melt periods are interrupted by periods with colder weather and new snowfall amounts. We therefore also analyzed the maximum difference in the relative part of the snowpack that consists of melt forms in the observed profiles, between two subsequent snow profiles. This is an indication of the progress of the melt water front inside the snowpack. However, also this did not reveal any statistically significant information. We think that ultimately, there are many factors contributing to high or low  $r^2$  values or good or poor estimates of the arrival date: warming rate of the snowpack, presence of capillary barriers and ice layers that may trigger preferential flow, errors in meteorological measurements and errors in snow lysimeter measurements. The snow lysimeter at the Weissfluhjoch measurement site has a surface area of 5  $m^2$ . As an illustration, Fig. 2 in Kattelmann (2000) shows that with this size, the variation coefficient is still quite large, and that an area of at least 10  $m^2$  may be required to more accurately capture snowpack runoff. Furthermore, inconsistencies in the bi-weekly profiles are present due to the subjective component of judging grain size and shape, as well as the fact that multiple observers are responsible for the snow profiles. All these factors make the analysis of factors contributing to high or low model performance difficult. We provide a discussion now in the revised manuscript, see P11, L30 to P12, L3.

8. P10-11 In the discussion section, descriptions '(1) P10L27-31, the absence of studies at low input rates makes the general validity of condition 1 we implemented uncertain (2) P10L6-9 Muti-dimensional snowpack models may help here to develop better understanding of the heat exchange processes between preferential flow paths and surrounding snow matrix, as a function of the number density of active preferential flow paths' are important messages. These suggestions provide the idea for valuable laboratory experiment and analysis using other model. If authors have other idea (e.g. the experiment to parameterize the process of transition from preferential flow to matrix flow.) and added in the manuscript, it will be welcomed as valuable information for reader studying wet snow.

We think the most crucial step is to investigate and quantify the heat exchange between preferential flow paths and the surrounding ice matrix, as a function of preferential flow area and the number of preferential flow paths. We expect that the vertical percolation speed of the preferential flow fingers is slowed down when percolating cold snow where considerable refreeze may take place. We think that a deeper investigation requires laboratory experiments with low water input rates (this may be difficult to achieve), high resolution temperature measurements (probably using infra-red photography) as well as dye-tracer to follow the wetting of the snowpack. Related to point 2, larger sample sizes may be required to prevent preferential flow finger width exceeding the sample size. We additionally think that using multi-dimensional models to simulate laboratory experiments helps to quantify the heat and water flow and to verify to what extent formulations of the heat and water exchange between preferential flow paths and the surrounding matrix reproduce the observations in laboratory experiments. For example, when a preferential flow path hits a microstructural transition, it starts spreading over the interface at some point. This is often explained via the water entry suction, yet the exceedance of the water entry suction in the preferential flow path in our simulations was not occurring often enough to successfully reproduce ice layers. Detailed laboratory and numerical studies using multi-dimensional models may assess the liquid water content distribution inside a preferential flow path. However, here it is important to note that some studies suggest that the water flow and wetness distribution inside a preferential flow path can probably not be described by Richards equation (DiCarlo, 2013). Ultimately, formulations for heat and water exchange between preferential flow paths and the surrounding matrix can then be incorporated in a model framework for 1-dimensional snowpack models, such as the one which we propose in this study. We amended the manuscript at this point, see P12, L30 to P13, L5.

**Response to Reviewer 2**

The authors have chosen to address a challenging topic and the results of this effort reflect the difficulty of simulating a heterogeneous, three dimensional, multi-process phenomenon in a 1-dimensional framework. While there is certainly much progress and future work to come in this area, the authors have presented a valuable analysis and initial framework that can be adapted and built upon in the future. As such, I am recommending that this work be accepted for publication in The Cryosphere, following some minor revisions that I believe will improve the quality and usefulness of the work.

We thank the reviewer for his constructive comments on the manuscript. Please find our detailed response to the issues raised by the reviewer below.

**Major comment**

Much of the framework relies on the interpretation of the results of Katsushima et al. (2013) as interpreted by Hirashima et al. (2014). They found that when water had reached the bottom of the profiles in their laboratory experiments, that the fractional area at 4 cm depth was smaller with larger snow grain size. The wetted fraction was interpreted here as the preferential flow fraction. As grain size decreased the wetted fraction increased and this was interpreted as the preferential flow fraction being larger. However, as this extends to the finest grain size, it is stated that no preferential flow was observed, and indeed if slower matric flow was happening, the deepest wetted area would be larger and more uniform. If that is what happened, and matric flow had extended beyond the 4 cm depth, then we have no clear transition defined between the two flow regimes. I would like the authors to discuss the interpretation of the experiment on which Figure 2 is based.

We partly agree with the interpretation of the experiments by the reviewer. We think that at some point, it may become not possible anymore to clearly define matrix flow and preferential flow regimes. If one observes that 90% of the snowpack is wetted during infiltration, could you still call it preferential flow and flow fingering, or is it just matrix flow where 10% of the snowpack is somehow not involved in matrix flow? However, we should also consider the possibility that the observations in the studies by Katsushima et al. (2013) and Hirashima et al. (2014) are not sufficient to conclude that both regimes cannot be clearly defined. It may happen that the finger width exceeds the size of the container used in the experiments such that the preferential flow finger is not identifiable as such anymore DiCarlo (2013). The diameter of the rings used in the experiments by Katsushima et al. (2013) was 5 cm and larger rings may be required to study water flow for small grains. On the other hand, in soil science, experiments also have demonstrated that for certain flow regimes and small grain size, no preferential flow forms and the flow is considered stable DiCarlo (2013). This was also found for larger grain sizes at low infiltration rates. We think that at this point, it is uncertain whether for fine snow grains, a regime exists were no preferential flow forms, or that preferential flow in snow forms under any circumstance for infiltration in snow. We hope that future experimental studies will address these issues. We did not explicitly

mention in the manuscript the absence of preferential flow for fine grains in the observations by Katsushima et al. (2013) when discussing Fig. 2. We now report this on P4, L2-3. However, more importantly, the smallest grain size class used in the experiments by Katsushima et al. (2013) is not often found in the SNOWPACK simulations. This grain size class is mostly associated with new snow fall and subsequent metamorphism generally quickly increases the grain size. We amended the manuscript with an extra figure (see Fig. 6 in the revised manuscript). In subfigure 6(e), the grain radius from the simulations is shown. Here, we coloured the grain radius black when it is below 0.16 mm, which corresponds to the lowest grain size class used by Katsushima et al. (2013). As can be seen, the important parts of the snowpack involved in preferential flow have a grain radius larger than the specific grain size class for which no preferential flow was observed.

**Specific Comments:**

• P2 line 15: Although I would not expect a through presentation of the work of Colbeck (1979) or Marsh and Woo (1985), the authors have stated that these works were not widely adopted. A couple of sentences summarizing the main concepts presented in these studies, and any weaknesses that may have resulted in their lack of adoption, would help to inform the readers about the need for progress in this area of research.

We revised the manuscript as follows (see P2, L14-21 of the revised manuscript):

"In early attempts by Colbeck (1979) and Marsh and Woo (1985) to describe preferential flow in snowpack models, the water flow in snow is described as a flow in multiple flow paths. In Colbeck (1979), flow paths are defined that differ in size and snowpack properties, which results in different percolation speeds in the individual snow paths when applying Darcy's law. In Marsh and Woo (1985), the snowpack is divided in flow paths of equal size and snowpack properties, but based on comparted lysimeter measurements, it is determined how much of the total flux is transported in each of the individual flow paths. Both approaches never found widespread adoption, probably because they require a-priori specification of the flow path variability (Marsh, 1999)."

Why exactly the adoption of those approaches was limited is difficult to assess. It may be related to the uncertainty of the additional parameters needed for those descriptions. It may also be that most model developments focussed on processes that required more urgent attention. When taking the SNOWPACK model as an example: the model was originally developed for avalanche warning purposes and was later developed to study other areas, like snow hydrology, catchment hydrology, Antarctic snow covers, the Greenland ice sheet, etc. Furthermore, there seems to be a strong increase in interest in liquid water flow in snow in the recent years. The availability of high quality laboratory experiments, new modelling techniques, etc., turned the attention again to preferential flow. We therefore do not want to interpret too much why the early modelling attempts did not find widespread adoption.

• P2 lines 19-21: I suspect that the specific application suggested in the example would require at least a 2-d model. The dual-domain approach is a good starting place. Given that the authors point to three flow regimes (matric flow, flow fingering and macropore flow), it would seem appropriate to include multiple flow domains in the future; one step at a time.

Thank you for the comments. We actually think that the application for wet snow avalanche prediction may already benefit from the dual domain approach, not necessarily requiring a full 2-dimensional model. Here, acquiring accurate snow depth and meteorological forcing in avalanche slopes is probably a more important source of uncertainty than the liquid water flow modelling. We agree that potentially, the dual domain could be extended with additional domains for other water transport processes. However, we think that validation and calibration of the model will become increasingly difficult given the typical inaccuracies in forcing data (meteorological measurements) and validation data (snow lysimeter data) for snow models as well as deficiencies in model process descriptions.

- P7: I suspect that there would be an interaction between the layer thicknesses (resolution) and the threshold of defining ice lenses / layers. A thicker layer would require more water equivalent to reach a given threshold density, and this would be harder to achieve in terms of water and energy transfer. A higher resolution or finer layers should enable higher threshold density values for defining ice layers. *Indeed, this is illustrated in Fig. 7 in the original manuscript, Fig. 8 in the revised one. For a specific threshold, the probability of detection is higher using thinner layers, which also means that for the same probability of detection, the threshold can be set higher for thin layers than for thicker layers. However, our interpretation of this figure is still that the difference between high and normal resolution simulations is much smaller than the difference between observations and simulations. Many discrepancies between observed and simulated snow profiles in terms of layering are probably not caused by the initial layer thickness, but also by errors in meteorological forcing as well as model simplifications and model process representation errors.*
- P8 lines 7-17: This is an honest discussion of the performance issues but also shows that this methodology may be useful for future development. *Thank you.*
- P9 lines 1-5: Was the precipitation type generally known in the observations or was this 2003 event a specific example of a known case in which the model misclassified the precipitation type? In any event, if it was known that the precipitation was rain, the model's diagnosis could be overwritten for this event, and if precipitation type was generally known, the model could be fed rainfall and snowfall separately. It would remove a source of uncertainty from the results. If the precipitation type was not generally known, but merely appeared to follow expected patterns save for the 2003 example, then it is not necessary to make any changes, given the size of the dataset.

Precipitation type is generally not known for the Weissfluhjoch measurement series, and the separation in rain and snow is done on the basis of air temperature. However, this particular event is a known case where rain on cold snow formed an ice layer which sustained throughout the snow season. We did experiments to force the model with rainfall for this period, but still the ice layer did not form. Probably the following reasons play a role: the SNOWPACK model is mostly run in 15 minute time steps and processes are solved sequentially. This means that at the beginning of the time step, new snow is added to the domain. Then the heat equation is solved to describe the temperature change over the 15 minute time step. Excess energy present after 15 min. is converted into snow melt, or, vice versa, in refreeze. Then, water flow is calculated over these 15 minutes. The fact that the equations are currently not solved in a coupled way but in a sequential way leads to the situation where water from rainfall can percolate out of the surface layer before there is time to refreeze the water. Furthermore, the water from rainfall is put in a layer of about 1.5 cm. In reality, one can imagine that rain drops hitting a snow surface below freezing will freeze directly at the surface, thereby creating an ice layer. It may even be the case that the ice layer thickens because the pore space at the surface vanishes, creating ponding conditions. Radiative cooling or rain with near surface air temperatures slightly below  $0^{\circ}C$  may cause this ponding water to refreeze. If one really wants to force the model to form an ice layer when rain falls on a cold snowpack, we can think of a mechanism where the upper most layer of the snowpack is split and the upper part of the splitted layer is forced to take ice density. On P9, L30 to P10, L3 of the revised manuscript, we report and discuss now that even forcing the precipitation to be rainfall did not create an ice layer in the

simulations.

- P9 lines 7-13: I would expect the spread in simulated snow density to be somewhat smaller than observed with such a model, based on the fact that snow pits sample small spatial areas, and the presence of discontinuous ice layers and fingering may increase the number of samples necessary to obtain a reliable estimate of the mean density and its variability at a given depth. The PFP simulations in Figure 6 appears to capture the density distribution slightly better than the Richards equation alone. Have other statistical measures been explored as alternatives or in addition to r2? A comparison of the mean and variance may show closer agreement with the PFP simulations. There is Willmot's index of agreement or one of the revised formulations. The objective is not to choose the statistic that makes the model appear better, but r2 has been criticized as being insensitive to important factors of model performance.
  - As snow density in the field is generally measured in sections of about 20-30cm depth, the presence of ice layers, with typical vertical extents of less than 2 cm, is hardly detectable in the measurements. We think that simulations with preferential flow do have a slightly lower performance in reproducing snow density in the simulations, for the reasons mentioned in P10L14-18 in the revised manuscript. We now also tested using Willmott's index and also with this index, simulations with preferential flow perform slightly worse on overall density distribution (the index decreases from 0.84 for simulations with Richards equation only to 0.83 for simulations with preferential flow). However, it seems that when changing from the 20-30 cm thick snow segments with measured density to high resolution vertical spacing, the simulations with preferential flow may better capture the variability by reproducing the ice layers. Measuring ice layer density is complicated (Watts et al., 2016) and we currently lack a good dataset of high resolution snow density measurements to validate the density distribution at the vertical resolution of the simulations (ca. 2 cm). Currently ongoing field campaigns using snow micro penetrometry (SMP, Schneebeli and Johnson (1998)) may change this situation soon and offers future potential to more in-depth verify the snow microstructure and snow density in the simulations in high detail. We report the results from the statistical test of Willmott's index in the revised manuscript on P10, L11-13. Additional discussion about this point is provided on P10, L13-20. Potential future measurements are discussed now on P10, L21-24 of the revised manuscript.
- In looking at Figures 8 and 9, there are differences in the performance of the REQ and REQ+PF models from year to year. It would be interesting to compare the conditions against a ranking of differences in r2 (Fig. 8) and arrival date (Fig. 9). Are there specific snowpack or meteorological conditions that are correlated with the differences in performance between years? Knowledge of this may be useful for future model development.

We provide here the same response as to Reviewer 1, who raises a similar issue: it certainly is an interesting suggestion to try to explain year-to-year differences in performance of snowpack runoff simulations. However, we did not find a statistically significant correlation between, for example, the  $r^2$  value or arrival date and the number of ice layers or the number of jumps in grain size or hardness observed in the profiles. We tested both linear correlation using Pearson correlation as well as rank correlation coefficients (Spearman, Kendall). One issue is that, as Reviewer 2 points out, meteorological conditions also vary from year to year. In some melt seasons, percolation speeds through the snowpack are high, in others it takes a few weeks for the whole snowpack to become wet. These differences arise from weather patterns: in some years warm weather prevails for several weeks in the melt season, leading to a quick wetting of the snowpack, whereas in other years, snow melt periods are interrupted by periods with colder weather and new snowfall amounts. We therefore also analyzed the maximum difference in the relative part of the snowpack that consists of melt forms in the observed profiles, between two subsequent snow profiles. This is an indication of the progress of the melt water front inside the snowpack. However, also this did not reveal any statistically significant information. We think that ultimately, there are many factors contributing to high or low  $r^2$  values or good or poor estimates of the arrival date: warming rate of the snowpack, presence of capillary barriers and ice layers that may trigger preferential flow, errors in meteorological measurements and errors in snow lysimeter measurements. The snow lysimeter at the Weissfluhjoch measurement site has a surface area of  $5 m^2$ . As an illustration, Fig. 2 in Kattelmann (2000) shows that with this size, the variation coefficient is still quite large, and that an area of at least  $10 m^2$  may be required to more accurately capture snowpack runoff. Furthermore, inconsistencies in the bi-weekly profiles are present due to the subjective component of judging grain size and shape, as well as the fact that multiple observers are responsible for the snow profiles. All these factors make the analysis of factors contributing to high or low model performance difficult. We provide a discussion now in the revised manuscript, see P11, L30 to P12, L3.

**Technical Comments:**

- P1 line 18: Change 'extend' to 'extent' *Changed, see P1, L18. Thank you.*
- P1 line 20-23: Change 'Water may flow... instead (Eiriksson et al., 2013)' to 'Water may flow laterally over ice layers or crusts, which reduces travel times and has a significant impact on catchment scale hydrology; alternatively, preferential flow in snow may promote vertical percolation instead (Eiriksson et al., 2013)'.

Changed, see P1, L20-23, thank you.

- P1 line 23: This reads as if the reader is familiar with the increased melt on the Greenland Ice Sheet, which is likely to be true but, I would add a word or two to indicate the time frame. *The reference we used to support this claim investigated a period of about 2 decades to relate changes in firn structure to warming trends, although the process may be occurring over longer time scales. This is revised in the manuscript, see P1, L23.*
- P1 line 24: Change 'extends in' to 'extents, on'. *Changed, see P1, L24 to P2, L1, thank you.*
- P2 line 6 (possibly elsewhere): I find the use of the term 'snow covers' to be awkward. I suggest the use of the term 'snowpack' or 'snowpacks'.
   Changed throughout the manuscript, thank you.
- P2 line 15: I would change 'deployment' to 'adoption'. *Changed, see P2, L20, thank you.*
- P7 line 5: Change '... given the suspicion of problems...' to '... due to suspected problems...'. *Changed, see P7, L22, thank you.*
- P9 lines 8-9: Change '.... is well reproduced...' to '... is reproduced well...' *Changed, see P10, L6-7, thank you.*

$$h_{\rm we} = 0.0437(2r_{\rm g}) + 0.01074,.$$
(2)

One important condition to reach the water entry pressure is water ponding on a microstructural transition inside the snowpack (*Hirashima et al.*, 2014; *Avanzi et al.*, 2015). This is denoted with 2 in Fig. 1. To achieve the high LWC value observed in experiments (*Avanzi et al.*, 2015), we use the geometric average to calculate the hydraulic conductivity between snow layers (*Wever et al.*, 2015). In our implementation, the amount of water in the matrix part in excess of the threshold corresponding to the water entry pressure of the layer below, is moved to the preferential flow domain in the layer below (denoted with 3 in Fig. 1). If afterwards A capillary overshoot condition was found in snow (*Katsushima et al.*, 2013), which means that the capillary pressure in the ponding layer decreases again after preferential flow forms. This increases the liquid water content in the preferential flow paths and to mimic this effect, we allow more water to flow from matrix flow to preferential flow once the

5 threshold is exceeded than only the amount of water above the threshold. If after the water in excess of the threshold is moved and the saturation (i.e., ratio of water volume to pore volume) in the layer in the matrix domain is still higher than the saturation in the layer below in the preferential flow domain, the saturation is equalized by an equivalent water flow with the following approach. Equal saturation in a specific layer with index *i* in the matrix domain and a layer with index *j* in the preferential flow domain can be expressed as:

$$\quad \frac{\theta_{\rm m} - \theta_{\rm r,m}}{\theta_{\rm s,m} - \theta_{\rm r,m}} \frac{\theta_{\rm m}^{i} - \theta_{\rm r,m}^{i}}{\theta_{\rm s,m}^{i} - \theta_{\rm r,m}^{i}} = \frac{\theta_{\rm p} - \theta_{\rm r,p}}{\theta_{\rm s,p} - \theta_{\rm r,p}} \frac{\theta_{\rm p}^{j} - \theta_{\rm r,p}^{j}}{\theta_{\rm s,p}^{j} - \theta_{\rm r,p}^{j}},\tag{3}$$

Where the subscripts m and p denote the matrix and preferential flow domain, respectively,  $\theta$  is the LWC (m3 m-3),  $\theta_r$  is the residual LWC (m3 m-3) and  $\theta_s$  is the saturated LWC (m3 m-3). In the model, layers are counted from below, such that equalizing saturation between the matrix domain in the layer above and the preferential flow domain in the layer below corresponds to j = i - 1.

15 Given layer thicknesses  $\frac{L_{\text{m}}}{L_{\text{m}}} \frac{1}{L_{\text{m}}} \frac{L_{\text{p}}}{L_{\text{m}}} \frac{L_{\text{p}}}{L_{\text{m}}} \frac{L_{\text{p}}}{L_{\text{p}}}$  for the layers in the matrix flow and preferential flow domain, respectively, the total LWC in the matrix and preferential flow layer is defined as:

$$\theta_{\rm tot} = \theta^i{}_{\rm m}L^i{}_{\rm m} + \theta^j{}_{\rm p}L^j{}_{\rm p} \tag{4}$$

Under the requirement of an equal degree of saturation for a given total LWC, we can solve Eq. 3 for  $\theta_{\rm m}$ :  $\theta_{\rm m}^i$ :

$$\theta^{i}_{m} = - \underbrace{\frac{\left(\theta_{r,m}\theta_{s,p} - \theta_{r,p}\theta_{s,m}\right)L_{m} + \left(\theta_{r,p} - \theta_{s,p}\right)\theta_{tot}}{\left(\theta_{s,m} - \theta_{r,m}\right)L_{m} + \left(\theta_{s,p} - \theta_{r,p}\right)L_{p}}}_{\left(\theta^{i}_{s,m} - \theta^{i}_{r,m}\right)L_{m}^{i} + \left(\theta^{j}_{s,p} - \theta^{j}_{s,p}\right)\theta_{tot}}_{\left(\theta^{i}_{s,m} - \theta^{i}_{r,m}\right)L_{m}^{i} + \left(\theta^{j}_{s,p} - \theta^{j}_{r,p}\right)L_{p}^{j}}},$$
(5)

20 after which  $\frac{\theta_{\rm p}}{\theta_{\rm p}} \frac{\theta_{\rm j}}{\theta_{\rm p}}$  can be found by applying Eq. 4.

[revised manuscript text omitted]
- 25 upper part of the snowpack only. The distribution of liquid water is showing that the preferential flow (Fig. 6b) is percolating ahead of the matrix flow (Fig. 6a). This partly is due to the absence of phase changes for water in preferential flow, but also that due to the lower area, and thereby lower value for  $\theta_s$ , such that hydraulic conductivity increases faster with increasing LWC. In contrast to matrix flow, preferential flow reaches areas where the snowpack is still below freezing (Fig. 6c).

Ponding at microstructural interfaces is occurring in both the matrix and the preferential flow domain<del>and it marks the layers</del> were water. In the example, a jump in snow density (Fig. 6d) and grain radius (Fig. 6e) around 165 cm and 210 cm in the

30 were water. In the example, a jump in snow density (Fig. 6d) and grain radius (Fig. 6e) around 165 cm and 210 cm in the snowpack mark the layers where water accumulates, refreezes and forms ice layers (Fig. 6d, f). Solving Richards equation twice (for both domains) appears to be able to identify those layers. Refreezing locally increases the snow temperature to melting temperature (Fig. 6c). Initially, the model identifies refreeze inside the snow layer and marks the layer as a melt-freeze crust. Once dry snow density exceeds 700 kg/m3, the layer is marked as an ice layer (Fig. 6f). Note that Fig. 6e shows that most of the

snowpack consists of grain sizes for which preferential flow was observed in the experiments by *Katsushima et al.* (2013). The smallest grain size class from those experiments, for which no preferential flow was observed, is only found in the new snow layers during snowfall (black coloured areas), after which metamorphism rapidly increases grain size to regimes for which preferential flow was observed. This justifies the application of Eq. 1 for the full range of grain size in the model.

- In addition to preferential flow, ice layers can also form by surface processes. For example, rainfall in November 2003 in a sub-freezing snow cover formed an ice layer at the surface and this ice layer was subsequently observed during the rest of the 2004 snow season (see Fig. S5 in the Online Supplement). This layer is not reproduced in the SNOWPACK model, as . Firstly, the model did not recognize the precipitation as rainfall due to the low air temperature during the event. This layer Second, even when the model was forced to interpret the precipitation as rainfall, the ice layer did not form at the surface. The model
- 10 solves for the heat and water flow sequentially in a 15 min. time step, whereas the formation of an ice layer during rainfall is occurring on shorter time scales. Furthermore, we hypothesize that rain droplets probably freeze directly upon contact with the snow surface, creating an ice layer locally at the surface, whereas the SNOWPACK model considers the rainfall as an incoming flux in the top layer. When the available energy for freezing is not sufficient to freeze the full depth of the top layer in the model, an ice layer is not formed. In reality, the surface ice layer is possibly even hindering water entry to deeper layers.
- 15 which may thicken the surface ice layer. This particular ice layer in 2004 has been excluded in further analysis.

**4.3 Density Profiles**

Fig. 7 shows the observed snow density distribution in all snow profiles from snow seasons 2000-2015, typically representing vertical sections of 20-50 cm and sometimes smaller sections. The distribution of snow density for in these sections is well reproduced reproduced well by the simulations, although the spread in simulated snow density is lower than the observed

- 20 spread. All simulations provide very similar snow density distributions. The  $r^2$  value between observed and simulated density in the measurement sections is highest ( $r^2$ =0.74) for the simulations with Richards equation only and in *Wever et al.* (2015), it is shown that the temporal evolution and vertical distribution of snow density is in good agreement with measured snow density. With preferential flow, the  $r^2$  value reduces to 0.71. This reduction in model performance when using preferential flow is also confirmed when using Willmott's index of agreement (*Zambrano-Bigiarini*, 2014; *Willmott*, 1981), which was determined to
- 25 be 0.84 and 0.83 for simulations with Richards equation only and simulations with preferential flow, respectively. Nevertheless, the simulations with preferential flow are maintaining the overall snowpack density profile generally well and the reduction in  $r^2$  value may be attributed to the fact that calibration of snow settling functions was not performed considering the preferential flow model. Another reason may be that with preferential flow, more water is moved downward and less water can refreeze in matrix flow in the upper snow layers. It may be argued that an underestimation of snow settling can be compensated for by an
- 30 overestimation of refreezing water. In any case, the simulations with preferential flow stand out when looking at the highest snow density simulated in a layer within  $\pm 20$  cm of an observed ice layer. In this case, much higher snow densities are found in individual layers under consideration of preferential flow.

As the manual snow density measurements in the field represent much larger vertical sections (20-30 cm), these measurements cannot be used to verify the much higher resolution (1-2 cm or less) simulated densities on that scale. Time series using other

measurement techniques, as for example snow micro penetrometry (*Schneebeli and Johnson*, 1998) or measuring volume and mass of excavated ice layers (*Watts et al.*, 2016), may assist in a more in-depth model verification in the future.

[revised manuscript text omitted]

5 is overestimated in the simulations. In *Würzer et al.* (2016b) additional confirmation is provided that the onset of snowpack runoff during In *Würzer et al.* (2016b) additional analysis of the role of preferential flow in producing snowpack runoff during rain-on-snow events is better reproduced by the dual domain approachevents shows that for these events, snowpack runoff is better reproduced using the dual domain approach.

Altough considering preferential flow improved the snowpack runoff simulation, year-to-year variability in model performance

- 10 is still large. The difference between simulations with or without preferential flow are often smaller than the year-to-year variability. For example, in melt season 2001 and 2010,  $r^2$  values are low in both simulations and the difference between the simulations is smaller than the difference in  $r^2$  values with other years. Explanatory factors for the year-to-year variability in model performance for reproducing snowpack runoff were not found. Snowpack characterizing statistics, for example the observed number of ice layers or observed number of jumps in grain size and hardness, did not correlate significantly with  $r^2$
- 15 for snowpack runoff or the arrival date. This is probably due to a combination of errors in meteorological forcing conditions, observer bias in the bi-weekly snow profiles, and the limited representativeness of the snow lysimeter. Its surface area of  $5 \text{ m}^2$  may be considered too small to capture a representative area for snowpack runoff, such that randomness in the exact location where preferential flow paths form may influence the measurements (*Kattelmann*, 2000). Separating the individual errors appears to be difficult.

**20 5 Discussion**

[revised manuscript text omitted]

- 20 lacking formation of dense layers by the model. Around 20 % of observed ice layers in the field over 16 snow seasons were correctly simulated by the model in the form of a layer exceeding a dry snow density of 700 kg m-3. We showed that a dual domain approach is able to provide a physics based description of preferential flow and ice layer formation that is corresponding to findings in laboratory and field experiments. However, the formulation has two parameters that were calibrated for this study. Although we do not resolve individual flow paths, as is done in multi-dimensional snowpack models, a dual domain approach
- 25 can quantify the net effect of preferential flow on a snowpack in a 1-D snowpack model with much lower computational costs than multi-dimensional models and only marginally larger computational cost compared to 1-D non-multidomain models.

Acknowledgements. Funding for this research was provided by the Swiss National Science Foundation (NSF), grant number 200021E-160667. We thank the many employees of the WSL Institute for Snow and Avalanche Research SLF involved in taking the bi-weekly snow profiles at the Weissfluhjoch. Meteorological driving data for the SNOWPACK model as well as the bi-weekly snow profiles are accessible via doi

[revised manuscript text omitted]

- Illangasekare, T. H., R. J. Walter, Jr., M. F. Meier, and W. T. Pfeffer (1990), Modeling of meltwater infiltration in subfreezing snow, Water Resour. Res., 26(5), 1001-1012, doi:10.1029/WR026i005p01001.
- Katsushima, T., T. Kumakura, and Y. Takeuchi (2009), A multiple snow layer model including a parameterization of vertical water channel process in snowpack, Cold Reg. Sci. Technol., 59(2-3), 143-151, doi:10.1016/j.coldregions.2009.09.002, international Snow Science Workshop (ISSW) 2008.
- Katsushima, T., S. Yamaguchi, T. Kumakura, and A. Sato (2013), Experimental analysis of preferential flow in dry snowpack, Cold Reg. Sci. Technol., 85, 206-216, doi:10.1016/j.coldregions.2012.09.012.

Kattelmann, R. (1985), Macropores in snowpacks of Sierra Nevada, Ann. Glaciol., 6, 272–273.

Kattelmann, R. (2000), Snowmelt lysimeters in the evaluation of snowmelt models, Ann. Glaciol., 31(1), 406-410, doi:10.3189/172756400781820048.

10

5

- Lehning, M., P. Bartelt, B. Brown, T. Russi, U. Stöckli, and M. Zimmerli (1999), SNOWPACK calculations for avalanche warning based upon a new network of weather and snow stations, Cold Reg. Sci. Technol., 30(1-3), 145-157, doi:10.1016/S0165-232X(99)00022-1.
- Lehning, M., P. Bartelt, B. Brown, C. Fierz, and P. Satyawali (2002a), A physical SNOWPACK model for the Swiss avalanche warning Part II: Snow microstructure, Cold Reg. Sci. Technol., 35(3), 147–167, doi:10.1016/S0165-232X(02)00073-3.
- 15 Lehning, M., P. Bartelt, B. Brown, and C. Fierz (2002b), A physical SNOWPACK model for the Swiss avalanche warning Part III: Meteorological forcing, thin layer formation and evaluation, Cold Reg. Sci. Technol., 35(3), 169–184, doi:10.1016/S0165-232X(02)00072-1.
  - Machguth, H., M. MacFerrin, D. van As, J. E. Box, C. Charalampidis, W. Colgan, R. S. Fausto, H. A. J. Meijer, E. Moslev-Thompson, and R. S. W. van de Wal (2016), Greenland meltwater storage in firn limited by near-surface ice formation, Nature Clim. Change, 6(4), 390-393, doi:10.1038/nclimate2899.
- Marsh, P. (1988), Flow fingers and ice columns in a cold snow cover, in Proceedings of the 56th Annual Western Snow Conference, Western 20 Snow Conference, Kalispell, Montana.
  - Marsh, P. (1999), Snowcover formation and melt: recent advances and future prospects, Hydrol. Proc., 13(14-15), 2117–2134, doi:10.1002/(SICI)1099-1085(199910)13:14/15<2117::AID-HYP869>3.0.CO;2-9.

Marsh, P., and M.-K. Woo (1984), Wetting front advance and freezing of meltwater within a snow cover: 2. A simulation model, Water

25 Resour. Res., 20(12), 1865-1874, doi:10.1029/WR020i012p01865.

- Marsh, P., and M.-K. Woo (1985), Meltwater movement in natural heterogeneous snow covers, Water Resour. Res., 21(11), 1710–1716, doi:10.1029/WR021i011p01710.
- McGurk, B. J., and P. Marsh (1995), Flow-finger continuity in serial thick-sections in a melting sierran snowpack, in *Biogeochemistry of* Seasonally Snow-Covered Catchments (Proceedings of a Boulder Symposium, July 1995), IAHS publ. no. 228.
- Pfeffer, W. T., and N. F. Humphrey (1998), Formation of ice layers by infiltration and refreezing of meltwater, Ann. Glaciol., 26, 83-91. 30 Pfeffer, W. T., M. F. Meier, and T. H. Illangasekare (1991), Retention of Greenland runoff by refreezing: Implications for projected future sea level change, J. Geophys. Res., 96(C12), 22,117-22,124, doi:10.1029/91JC02502.

Phillips, M., A. Haberkorn, D. Draebing, M. Krautblatter, H. Rhyner, and R. Kenner (2016), Seasonally intermittent water flow through deep fractures in an Alpine Rock Ridge: Gemsstock, Central Swiss Alps, Cold Reg. Sci. Technol., 125, 117-127,

- 35 doi:10.1016/j.coldregions.2016.02.010.
  - Rees, A., J. Lemmetyinen, C. Derksen, J. Pulliainen, and M. English (2010), Observed and modelled effects of ice lens formation on passive microwave brightness temperatures over snow covered tundra, *Remote Sens. Environ.*, 114(1), 116–126, doi:10.1016/j.rse.2009.08.013.

- Rössler, O., P. Froidevaux, U. Börst, R. Rickli, O. Martius, and R. Weingartner (2014), Retrospective analysis of a nonforecasted rain-on-snow flood in the Alps a matter of model limitations or unpredictable nature?, *Hydrol. Earth Syst. Sci.*, *18*(6), 2265–2285, doi:10.5194/hess-18-2265-2014.
- Roy, A., A. Royer, O. St-Jean-Rondeau, B. Montpetit, G. Picard, A. Mavrovic, N. Marchand, and A. Langlois (2016), Microwave snow emission modeling uncertainties in boreal and subarctic environments, *Cryosphere*, *10*(2), 623–638, doi:10.5194/tc-10-623-2016.
- Schneebeli, M. (1995), Development and stability of preferential flow paths in a layered snowpack, in *Biogeochemistry of Seasonally Snow-Covered Catchments (Proceedings of a Boulder Symposium July 1995)*, edited by K. Tonnessen, M. Williams, and M. Tranter, p. 89–95, AHS Publ. no. 228.

Schneebeli, M., and J. Johnson (1998), A constant-speed penetrometer for high-resolution snow stratigraphy, Ann. Glaciol., 26, 107–111.

- 10 Singh, P., G. Spitzbart, H. Hübl, and H. Weinmeister (1997), Hydrological response of snowpack under rain-on-snow events: a field study, J. Hydrol., 202(1–4), 1–20, doi:10.1016/S0022-1694(97)00004-8.
  - Surfleet, C. G., and D. Tullos (2013), Variability in effect of climate change on rain-on-snow peak flow events in a temperate climate, *J. Hydrol.*, 479, 24–34, doi:10.1016/j.jhydrol.2012.11.021.

Vikhamar-Schuler, D., I. Hanssen-Bauer, T. Schuler, S. Mathiesen, and M. Lehning (2013), Use of a multilayer snow model to assess grazing

15 conditions for reindeer, Ann. Glaciol., 54(62), 214–226, doi:10.3189/2013AoG62A306.

Watts, T., N. Rutter, P. Toose, C. Derksen, M. Sandells, and J. Woodward (2016), Brief communication: Improved measurement of ice layer density in seasonal snowpacks, *Cryosphere*, 10(5), 2069–2074, doi:10.5194/tc-10-2069-2016.

Wever, N., C. Fierz, C. Mitterer, H. Hirashima, and M. Lehning (2014a), Solving Richards Equation for snow improves snowpack meltwater runoff estimations in detailed multi-layer snowpack model, *Cryosphere*, *8*(1), 257–274, doi:10.5194/tc-8-257-2014.

- 20 Wever, N., T. Jonas, C. Fierz, and M. Lehning (2014b), Model simulations of the modulating effect of the snow cover in a rain-on-snow event, *Hydrol. Earth Syst. Sci.*, 18(11), 4657–4669, doi:10.5194/hess-18-4657-2014.
  - Wever, N., L. Schmid, A. Heilig, O. Eisen, C. Fierz, and M. Lehning (2015), Verification of the multi-layer SNOWPACK model with different water transport schemes, *Cryosphere*, 9(6), 2271–2293, doi:10.5194/tc-9-2271-2015.

Wever, N., C. Vera Valero, and C. Fierz (2016), Assessing wet snow avalanche activity using detailed physics based snowpack simulations,

25 Geophys. Res. Lett., 43(11), 5732–5740, doi:10.1002/2016GL068428.

5

Williams, M. W., M. Rikkers, and W. T. Pfeffer (2000), Ice ccolumn and frozen rills in a warm snowpack, Green Lakes Valley, Colorado, U.S.A., Nord. Hydrol., 31(3), 169–186.

Willmott, C. J. (1981), On the validation of models, *Phys. Geogr.*, 2(2), 184–194, doi:10.1080/02723646.1981.10642213.

- WSL Institute for Snow and Avalanche Research SLF (2015), Manual bi-weekly snow profiles from Weissfluhjoch, Davos, Switzerland,
  doi:10.16904/2.
  - WSL Institute for Snow and Avalanche Research SLF (2015-09-29), Meteorological and snowpack measurements from Weissfluhjoch, Davos, Switzerland, 1, doi:10.16904/1, dataset.

Würzer, S., T. Jonas, N. Wever, and M. Lehning (2016a), Influence of initial snowpack properties on runoff formation during rain-on-snow events, *J. Hydrometeor*, *17*(6), 1801–1815, doi:10.1175/JHM-D-15-0181.1.

- 35 Würzer, S., N. Wever, R. Juras, M. Lehning, and T. Jonas (2016b), Modeling liquid water transport in snow under rain-on-snow conditions considering preferential flow, *Hydrol. Earth Syst. Sci. Discuss.*, doi:10.5194/hess-2016-351.
  - Yamaguchi, S., K. Watanabe, T. Katsushima, A. Sato, and T. Kumakura (2012), Dependence of the water retention curve of snow on snow characteristics, *Ann. Glaciol.*, 53(61), 6–12, doi:10.3189/2012AoG61A001.

- Ye, H., D. Yang, and D. Robinson (2008), Winter rain on snow and its association with air temperature in northern Eurasia, *Hydrol. Proc.*, 22(15), 2728–2736, doi:10.1002/hyp.7094.
- Zambrano-Bigiarini, M. (2014), *hydroGOF: Goodness-of-fit functions for comparison of simulated and observed hydrological time series*, r package version 0.3-8, https://CRAN.R-project.org/package=hydroGOF.